# STRUCTURE-ALIGNED PROTEIN LANGUAGE MODEL

## ABSTRACT

Protein language models (pLMs) pre-trained on vast protein sequence databases excel at various downstream tasks but often lack the structural knowledge essential for some biological applications. To address this, we introduce a method to enrich pLMs with structural knowledge by leveraging pre-trained protein graph neural networks (pGNNs). First, a latent-level contrastive learning task aligns residue representations from pLMs with those from pGNNs across multiple proteins, injecting inter-protein structural information. Additionally, a physical-level task integrates intra-protein information by training pLMs to predict structure tokens. Together, the proposed dual-task framework effectively incorporates both inter- and intra-protein structural knowledge into pLMs. Given the variability in the quality of protein structures in PDB, we further introduce a residue loss selection module that uses a small model trained on high-quality structures to select reliable yet challenging residue losses for the pLM to learn. Applying our structure alignment method as a simple, lightweight post-training step to the state-of-the-art ESM2 and AMPLIFY yields notable performance gains. These improvements are consistent across a wide range of tasks, including substantial gains in deep mutational scanning (DMS) fitness prediction and a 59% increase in P@L for ESM2 650M contact prediction on CASP16. Furthermore, we demonstrate that these performance gains are robust, scaling with model sizes from 8M to 650M and extending to different downstream tasks. The data, code, and resulting SaESM2 and SaAMPLIFY models will be made publicly available upon publication.

## 1 INTRODUCTION

Building on recent progress in natural language processing (Brown et al., 2020; Devlin et al., 2019), researchers have focused on pre-training protein language models (pLMs) on vast databases of protein sequences with masked language modeling (Rives et al., 2019; Hayes et al., 2024; Fournier et al., 2024) and next token prediction (Ferruz et al., 2022). These pLMs learn representations that researchers have demonstrated hold substantial potential across a variety of biological applications, including protein function annotation, enzyme-catalyzed reaction prediction, and protein classification (Hu et al., 2022).

Additionally, Rives et al. (2019) observed that structural information emerged in the models' latent representations without supervision. Nonetheless, while the sequence-only nature of pLMs contributes to their widespread adoption, they often struggle in tasks requiring detailed structural insights. For instance, the structured informed ESM-GearNet outperforms ESM2 by $9.7\%$ on the Human Protein-Protein Interaction classification task (Xu et al., 2022; Su et al., 2024). In this paper, we aim to develop a pLM that preserves its sequence-only nature for broader applicability yet is augmented with structural insights.

Given the availability of open-source pre-trained protein graph neural networks (pGNNs) (Zhang et al., 2023; Chen et al., 2023; Jumper et al., 2021), we investigate integrating pGNN-derived structural insights into pLMs. Specifically, we introduce a latent-level contrastive learning task for the structural alignment of pLMs. As illustrated in Figure 1, this task aligns residue hidden representations from the pLM ($\boldsymbol{h}_a$) with those from the pGNN ($\boldsymbol{h}_g$) across a batch of $B$ proteins. During this process, the pGNN is frozen while the pLM is optimized to minimize the contrastive learning

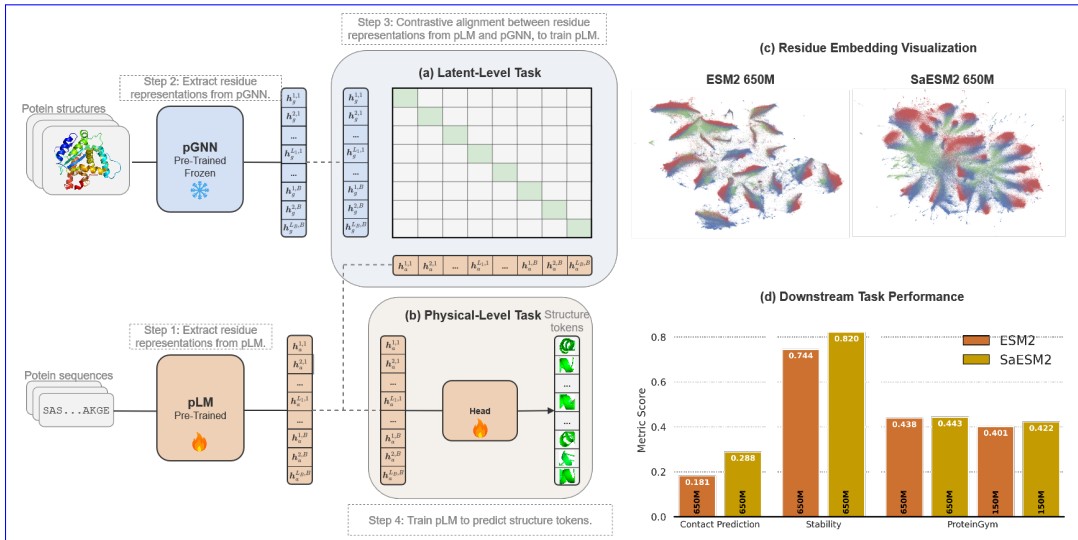

Figure 1: Overview of the *dual-task framework*. (a) Latent-level task: contrastively aligns residue representations from the pLM and pGNN, allowing the pLM to learn inter-protein structural knowledge. (b) Physical-level task: trains the pLM to predict structural tokens, incorporating intra-protein knowledge. (c) Residue embedding visualization: UMAP colored by secondary structure, showing that alignment improves separation. (d) Downstream task performance: structural knowledge improves contact map prediction, thermostability estimation, and fitness landscape modeling.

loss, enriching the pLM with inter-protein[1] structural knowledge. However, pure contrastive alignment may overemphasize differing residue-level patterns across the dataset, neglecting intra-protein[1] structural context (Zheng & Li, 2024). To address this, we add a physical-level task that trains the pLM to predict structural tokens $z$ (representing physical conformations (van Kempen et al., 2022)) from its residue representations $h_a$. This reinforces the encoding of each residue within its protein, thereby enriching the pLM with intra-protein structural knowledge.

We combine latent and physical tasks, yielding three residue loss types for a batch of proteins with a total length $N$: (i) $N$ sequence-to-structure contrastive losses from the latent-level task, (ii) $N$ structure-to-sequence contrastive losses from the latent-level task, and (iii) $N$ structure token prediction losses from the physical-level task. The *dual-task framework* effectively integrates inter-protein and intra-protein residue-level structural knowledge (§3.1). The masked language modeling loss is additionally incorporated to preserve the sequential knowledge of pLMs.

Given that some protein structure regions in the PDB are ambiguous or inaccurate (Burley et al., 2019), we propose a *residue loss selection* module that prioritizes residue losses aligned with high-quality protein structures across the $3 \times N$ total residue losses (§3.2). First, we use resolution and R-free metrics (Morris et al., 1992) to curate a high-quality reference set and train a small reference model on the set. Next, we compute the *excess loss*, defined as the difference between the residue loss of the current model and that of the reference model (Mindermann et al., 2022). Residue losses with high excess loss are selectively included in each loss type as they exhibit greater learnable potential. This module filters out inaccurate residues with high reference loss and easy residues with low current loss. By focusing on challenging yet reliable residue losses, the module improves both training effectiveness and efficiency.

We conducted 10 ablations to validate our design choices. Our analysis demonstrates that the proposed *dual-task framework* improves performance, with *residue loss selection* providing further gains. The models were evaluated on a comprehensive suite of benchmarks. We assess perfor-

---

[1] Note that "inter-protein" and "intra-protein" refer to tasks involving multiple proteins and within a single protein, respectively. This usage differs from the biological definition, where "inter-protein" refers to interactions between two proteins, and "intra-protein" refers to interactions within a single protein chain.

mance on deep mutational scanning (DMS) fitness prediction using ProteinGym (Notin et al., 2023) and on direct structural validation using a contact prediction task on withheld data from CASP16 (Yuan et al., 2025). We further test generalization on 9 tasks from xTrimoPGLM (Chen et al., 2024) and 9 from SaProt (Su et al., 2023), and evaluate language modeling fidelity using pseudo-perplexity on the high-quality held-out validation set from Fournier et al. (2024). We find that structure alignment is a computationally lightweight post-training step that, depending on the downstream task, either matches or exceeds the performance of the original model.

To summarize, our contributions are three-fold:

- We propose a *dual-task framework* that integrates inter-protein and intra-protein residue-level structural knowledge into pLMs, while fully retaining their language modeling capabilities.
- We develop a *residue loss selection* module that prioritizes challenging yet reliable residue losses, enhancing the learning process of pLMs.
- We conduct extensive experiments that demonstrate the effectiveness of our method across model sizes, model families and downstream tasks.

## 2 PRELIMINARIES

In this section, we introduce the preliminaries of protein language models, structure embeddings, and structure tokens used in this study, and provide a more detailed review of related work in Appendix B.

### 2.1 PROTEIN LANGUAGE MODELS

Proteins can be represented as sequences of amino acids, where each amino acid $a_i$ belongs to the set of 20 common types. A protein sequence of length $L$ is denoted as $\boldsymbol{a} = (a_1, a_2, \ldots, a_L)$.

Protein language models are pre-trained on hundreds of millions of protein sequences using objectives such as masked language modeling (MLM) (Hayes et al., 2024; Rives et al., 2019) and next-token prediction (Ferruz et al., 2022), capturing rich biophysical information. In this study, we focus on MLM-based pLMs, as proteins are not intrinsically left-to-right, and MLM has been shown to be highly effective for downstream tasks (Lin et al., 2023b).

A pre-trained pLM parameterized by $\boldsymbol{\theta}$ is represented as $\mathrm{pLM}(\cdot; \boldsymbol{\theta})$. The latent representation of a protein sequence $\boldsymbol{a}$ is denoted as $\mathrm{pLM}(\boldsymbol{a}; \boldsymbol{\theta}) \in \mathbb{R}^{L \times D_a}$, where $D_a$ is the embedding dimension. During pre-training, a subset of positions $\mathcal{M} \subset \{1, \ldots, L\}$ is replaced with a [mask] token:

$$\tilde{a}_i = \begin{cases} [\text{mask}], & \text{if } i \in \mathcal{M}, \\ a_i, & \text{otherwise,} \end{cases} \tag{1}$$

where $\tilde{\boldsymbol{a}} = (\tilde{a}_1, \ldots, \tilde{a}_L)$ represents the modified sequence. The model is trained to reconstruct the masked tokens by minimizing the masked language modeling loss:

$$\mathcal{L}_{\mathrm{mlm}}(\boldsymbol{\theta}, \boldsymbol{\alpha}) = \frac{1}{|\mathcal{M}|} \sum_{i \in \mathcal{M}} \ell_{\mathrm{CE}}\Big(\mathrm{MLP}\big(\mathrm{pLM}(\tilde{\boldsymbol{a}}; \boldsymbol{\theta})_i; \boldsymbol{\alpha}\big), a_i\Big), \tag{2}$$

where $\ell_{\mathrm{CE}}$ is the cross-entropy loss, $\mathrm{pLM}(\boldsymbol{a}; \boldsymbol{\theta})_i \in \mathbb{R}^{D_a}$ is the embedding at position $i$, and $\mathrm{MLP}(\cdot; \boldsymbol{\alpha})$, parameterized by $\boldsymbol{\alpha}$, is a multi-layer perceptron head used during pre-training to predict the amino acid type.

### 2.2 PROTEIN STRUCTURE EMBEDDINGS

Protein language models generate residue-level embeddings from protein sequences. In addition to the sequence perspective, proteins exist as 3D structures, and this physical nature largely determines their biological functions. Recent studies have also investigated deriving residue-level embeddings directly from protein 3D structures. One approach is to use the residue-level hidden representations generated by AlphaFold2 (Jumper et al., 2021), although their effectiveness for downstream tasks has since been questioned (Hu et al., 2022). GearNet (Zhang et al., 2023) addresses this limitation by pre-training a protein graph model encoder using multiview contrastive learning. Similarly,

STEPS (Chen et al., 2023) improves protein structural representations by introducing multiple self-prediction tasks during graph model pre-training.

Given a protein graph $\boldsymbol{g}$, where each residue is a node and edges are defined based on both sequential and structural distances, a pre-trained protein GNN model outputs residue-level embeddings $\text{pGNN}(\boldsymbol{g}) \in \mathbb{R}^{L \times D_g}$, where $L$ is the number of residues, and $D_g$ is the embedding dimension of the graph-based residue representation.

### 2.3 Protein Structure Tokens

Inspired by the success of token-based protein language models, recent studies have explored the idea of tokenizing protein structures, representing a protein's 3D conformation as a series of discrete structure tokens. A protein structure with $L$ residues can be expressed as $\boldsymbol{z} = (z_1, z_2, \ldots, z_L)$, where $z_i$ denotes the structure token for the $i$-th residue.

Foldseek (van Kempen et al., 2022) introduces an efficient method for tokenizing protein structures, where each residue $i$ is described by its geometric conformation relative to its spatially closest residue $j$. While this approach has significantly accelerated homology detection, it incurs substantial information loss, thereby limiting its applicability to tasks requiring detailed structural reconstruction. To address this limitation, ProToken (Lin et al., 2023a) employs a symmetric encoder-decoder architecture that enables high-fidelity reconstruction of protein structures from tokens. Despite this advancement, these tokens have shown limited effectiveness in broader downstream applications (Zhang et al., 2024a).

Recently, Hayes et al. (2024) developed an effective vector quantization variational autoencoder (VQ-VAE) tokenizer and integrated structure and sequence into a multi-modal protein language model called ESM3. This approach effectively combines both modalities, improving the model's versatility. While we could not evaluate ESM3 ourselves due to licensing restrictions, we were able to retrieve its reported performance on the ProteinGym benchmark. AIDO (Zhang et al., 2024a) further enhances structure tokenization by introducing a novel VQ-VAE with an equivariant encoder and an invariant decoder, ensuring a more robust representation of protein structures.

## 3 Method

### 3.1 Dual-Task Framework

We present our *dual-task framework*, consisting of a latent-level task and a physical-level task.

**Latent-Level Task**   To incorporate structural insights from pre-trained pGNNs, we propose a latent-level contrastive learning task for the structure alignment of pLMs. Assuming a batch contains $B$ proteins, with a total of $N = \sum_{b=1}^{B} L_b$ residues, we perform contrastive learning across all residues. We denote the pLM hidden representation of the $i$-th residue from the $b_1$-th protein sequence $\boldsymbol{a}_{b_1}$ as $\text{pLM}(\boldsymbol{a}_{b_1}; \boldsymbol{\theta})_i$, and the pGNN hidden representation of the $j$-th residue from the $b_2$-th protein structure $\boldsymbol{g}_{b_2}$ as $\text{pGNN}(\boldsymbol{g}_{b_2})_j$. Note that the parametrization of the pGNN is omitted for brevity, as the pGNN is frozen during training while only the pLM parameters $\boldsymbol{\theta}$ are optimized.

To align these embeddings, we introduce two linear layers, $\boldsymbol{W}_a \in \mathbb{R}^{D_a \times D}$ and $\boldsymbol{W}_g \in \mathbb{R}^{D_g \times D}$, and a learnable scalar $s$, parameterized as $\boldsymbol{W} = [\boldsymbol{W}_a; \boldsymbol{W}_g; s]$, which project both embeddings into the same dimension $D$. The similarity score between residues is computed as:

$$\delta(i, b_1, j, b_2) = s\big(\text{pLM}(\boldsymbol{a}_{b_1}; \boldsymbol{\theta})_i \boldsymbol{W}_a\big)^{\top} \big(\text{pGNN}(\boldsymbol{g}_{b_2})_j \boldsymbol{W}_g\big), \qquad (3)$$

where $s$ follows the approach in CLIP (Radford et al., 2021). The contrastive step is similar to that of Robinson et al. (2023), with three notable exceptions: we keep the pGNN frozen, because our objective is to align the pLM, they discard the language modeling head, which is contrary to our final loss Equation 5 and they discard our physical level task (section 3.1), replacing it with another, higher-level, inter-protein contrastive loss. Their objective also seems to be more the improvement of protein sequential models, built in complete isolation from a folding task.

In our experiments, we primarily use GearNet (Zhang et al., 2023) as the pGNN, pre-trained on the AlphaFold2 database (Varadi et al., 2022). We also evaluated the Evoformer within AlphaFold2 (Jumper et al., 2021) but found GearNet embeddings to be more effective for our purpose.

The sequence-to-structure residue contrastive loss for the $i$-th residue in the $b_1$-th protein is:

$$\mathcal{L}_{\text{a2g}}(\boldsymbol{\theta}, \boldsymbol{W}, i, b_1) = -\log \frac{\exp\big(\delta(i, b_1, i, b_1)\big)}{\sum_{b_2=1}^{B} \sum_{j=1}^{L_{b_2}} \exp\big(\delta(i, b_1, j, b_2)\big)}. \tag{4}$$

The sequence-to-structure contrastive loss for the batch is then:

$$\mathcal{L}_{\text{a2g}}(\boldsymbol{\theta}, \boldsymbol{W}) = \frac{1}{N} \sum_{b_1=1}^{B} \sum_{i=1}^{L_{b_1}} \mathcal{L}_{\text{a2g}}(\boldsymbol{\theta}, \boldsymbol{W}, i, b_1). \tag{5}$$

A similar residue loss, $\mathcal{L}_{\text{g2a}}(\boldsymbol{\theta}, \boldsymbol{W}, b_2, j)$, can be defined for structure-to-sequence contrast, leading to the overall structure-to-sequence loss $\mathcal{L}_{\text{g2a}}(\boldsymbol{\theta}, \boldsymbol{W})$. The final latent-level loss is then given by:

$$\mathcal{L}_{\text{latent}}(\boldsymbol{\theta}, \boldsymbol{W}) = \frac{1}{2}\big(\mathcal{L}_{\text{a2g}}(\boldsymbol{\theta}, \boldsymbol{W}) + \mathcal{L}_{\text{g2a}}(\boldsymbol{\theta}, \boldsymbol{W})\big), \tag{6}$$

which enhances the pLM by integrating inter-protein residue-level structural knowledge.

**Physical-Level Task** However, pure contrastive alignment may overly emphasize residue-level structural patterns relative to the broader dataset, neglecting the intra-protein structural context. To address this, we introduce a physical-level task to reinforce the encoding of residue structure relative to its own protein.

This task trains the pLM to use the residue hidden representation to predict its structural token $\boldsymbol{z}$, which represents the residue's physical conformation (van Kempen et al., 2022). The structure token prediction loss for the $i$-th residue in the $b_1$-th protein is defined as:

$$\mathcal{L}_{\text{physical}}(\boldsymbol{\theta}, \boldsymbol{\beta}, i, b_1) = \ell_{\text{CE}}\Big(\text{MLP}\big(\text{pLM}(\boldsymbol{a}_{b_1}; \boldsymbol{\theta})_i; \boldsymbol{\beta}\big), z_{i,b_1}\Big), \tag{7}$$

where $\ell_{\text{CE}}$ denotes the cross-entropy loss, MLP represents a multi-layer perceptron, and $\boldsymbol{\beta}$ are the parameters of the MLP. The overall physical-level loss is given by:

$$\mathcal{L}_{\text{physical}}(\boldsymbol{\theta}, \boldsymbol{\beta}) = \frac{1}{N} \sum_{b_1=1}^{B} \sum_{i=1}^{L_{b_1}} \mathcal{L}_{\text{physical}}(\boldsymbol{\theta}, \boldsymbol{\beta}, i, b_1), \tag{8}$$

infusing the pLM with intra-protein residue-level structural knowledge.

**Overall Loss** In addition to the dual-task losses, we incorporate the original MLM loss to preserve the sequential knowledge of pLMs, resulting in the final loss function:

$$\mathcal{L}_{\text{overall}}(\boldsymbol{\theta}, \boldsymbol{\alpha}, \boldsymbol{W}, \boldsymbol{\beta}) = \mathcal{L}_{\text{mlm}}(\boldsymbol{\theta}, \boldsymbol{\alpha}) + \gamma_{\text{latent}} \mathcal{L}_{\text{latent}}(\boldsymbol{\theta}, \boldsymbol{W}) + \gamma_{\text{physical}} \mathcal{L}_{\text{physical}}(\boldsymbol{\theta}, \boldsymbol{\beta}), \tag{9}$$

where $\gamma_{\text{latent}}$ and $\gamma_{\text{physical}}$ are weighting factors set to $0.5$, ensuring equal importance for the latent-level and physical-level tasks. The weights are normalized such that $\gamma_{\text{latent}} + \gamma_{\text{physical}} = 1.0$, maintaining a balance between sequence and structure losses.

The proposed method is, to the best of our knowledge, the first to incorporate MLM regularization. Consequently, among similarly structure-aligned or contrastively fine-tuned sequence-only pLMs, we are the first to be evaluated on language modeling downstream tasks, such as Deep Mutation Scanning, which are crucial in drug discovery pipelines. Also, contrary to existing models such as SaProt (Su et al., 2024), there is no need for structure as input to enrich residue embeddings.This structure-agnostic capability is essential, given that proteins with intrinsically disordered regions, which lack a fixed tertiary structure, constitute a significant portion of the proteome. Independence from structural input ensures the model's applicability to any protein, including those with uncharacterized structures or those for which in silico folding predictions may be unreliable.

### 3.2 RESIDUE LOSS SELECTION

To address the challenge posed by ambiguous or inaccurate protein structures in the PDB (Burley et al., 2019), we propose a *residue loss selection* module. This module ensures both effectiveness and efficiency by prioritizing residue losses that align with high-quality protein structures.

**Reference Set**   We begin by curating a high-quality reference set using resolution and R-free metrics (Morris et al., 1992). Structures with resolution below $2.0\text{Å}$ and R-free lower than $0.20$ are selected as a clean reference set. We then train a smaller language model on the reference set with the same loss in Equation 9 and denote the optimized reference model parameters as $\boldsymbol{\theta}^r$, $\boldsymbol{\alpha}^r$, $\boldsymbol{W}^r$, $\boldsymbol{\beta}^r$. The resulting reference model is used to assess the residue loss of the alignment corpus.

**Excess Loss**   For each residue loss discussed in §3.1, we compute the *excess loss*, defined as the difference between the residue loss of the current model and that of the reference model:

$$\begin{aligned}
\mathcal{L}_{\text{a2g}}(\Delta, i, b_1) &= \mathcal{L}_{\text{a2g}}\big(\boldsymbol{\theta}, \boldsymbol{W}, i, b_1\big) - \mathcal{L}_{\text{a2g}}\big(\boldsymbol{\theta}^r, \boldsymbol{W}^r, i, b_1\big), \\
\mathcal{L}_{\text{g2a}}(\Delta, j, b_2) &= \mathcal{L}_{\text{g2a}}\big(\boldsymbol{\theta}, \boldsymbol{W}, j, b_2\big) - \mathcal{L}_{\text{g2a}}\big(\boldsymbol{\theta}^r, \boldsymbol{W}^r, j, b_2\big), \\
\mathcal{L}_{\text{physical}}(\Delta, i, b_1) &= \mathcal{L}_{\text{physical}}\big(\boldsymbol{\theta}, \boldsymbol{\beta}, i, b_1\big) - \mathcal{L}_{\text{physical}}\big(\boldsymbol{\theta}^r, \boldsymbol{\beta}^r, i, b_1\big).
\end{aligned} \tag{10}$$

where $\mathcal{L}_{\text{a2g}}(\Delta, i, b_1)$, $\mathcal{L}_{\text{g2a}}(\Delta, j, b_2)$, and $\mathcal{L}_{\text{physical}}(\Delta, i, b_1)$ represent the residue excess loss for sequence-to-structure, structure-to-sequence, and physical tasks, respectively.

**Loss Selection**   Residue losses with high excess loss are prioritized for inclusion in the training as they exhibit greater learnable potential. This effectively filters out inaccurate residues, which typically have high reference model loss, and excludes easy residues with low current model loss. We introduce a selection ratio $\rho$, selecting $N\rho$ residue losses for each type of loss. Taking $\mathcal{L}_{\text{a2g}}$ as an example, we rewrite Equation 5 as:

$$\mathcal{L}_{\text{a2g}}(\boldsymbol{\theta}, \boldsymbol{W}) = \frac{1}{N\rho} \sum_{b_1=1}^{B} \sum_{i=1}^{L_{b_1}} \mathbb{1}\big(\mathcal{L}_{\text{a2g}}(\Delta, i, b_1), \rho\big) \mathcal{L}_{\text{a2g}}\big(\boldsymbol{\theta}, \boldsymbol{W}, i, b_1\big), \tag{11}$$

where $\mathbb{1}\big(\mathcal{L}_{\text{a2g}}(\Delta, i, b_1), \rho\big)$ equals 1 if $\mathcal{L}_{\text{a2g}}(\Delta, i, b_1)$ ranks in the top $\rho$ of all $\mathcal{L}_{\text{a2g}}(\Delta, i, b_1)$ values, and 0 otherwise. This selection process is applied similarly for the other two types of losses. By focusing on challenging yet reliable residue losses, the *residue loss selection* module improves overall training effectiveness and efficiency.

## 4 EXPERIMENTS

### 4.1 STRUCTURE ALIGNMENT DETAILS

We aligned ESM2 and AMPLIFY using $129,732$ proteins from OpenFold (Ahdritz et al., 2023) present in the PDB database, of which $116,713$ are for training and $13,019$ for validation. We systematically verified that our sequences were deposited in the PDB in December 2021 to the latest. As a consequence, our training set is fully deduplicated against CASP16 , meaning CASP16 represents a gold standard test set for downstream evaluation of our models.

The training protocol is adapted from the AMPLIFY stage-2 configuration (Fournier et al., 2024) with several modifications. We extend the pre-training with 20 epochs on our alignment dataset, with the learning rate linearly warming up from 0 to the peak rate over the first two epochs, followed by a cosine decay schedule for the subsequent 18 epochs. The peak rate for the language model is set at $1 \times 10^{-4}$, as per the AMPLIFY standard, while other modules, such as the structural linear classifier and the contrastive learning module, are set at $1 \times 10^{-3}$. The selection ratio $\rho$ is set to $0.8$.

We employ the Zero Redundancy Optimizer (ZeRO) with DeepSpeed and use 8 H100 GPUs. The effective batch size is $4,096$ samples at a sequence length of $2,048$, with longer proteins being randomly truncated. Our post-training alignment method is particularly compute efficient, taking under 6 hours for the largest ESM2 model considered, and under 1 hour for the smallest model.

### 4.2 BASELINE MODELS

We evaluate the following sequence-only baseline pLMs: (1) **ESM2**: the standard ESM2 650M model (Lin et al., 2022); (2) **AMPLIFY**: the standard AMPLIFY 350M model (Fournier et al., 2024); (3) **ESM2-S**: a variant of ESM2 fine-tuned for fold classification (Zhang et al., 2024b); (4) **ISM**: a variant of ESM2 optimized for structure token prediction (Ouyang-Zhang et al., 2024); (5)

**S-PLM**: a different contrastive post-training method applied to ESM2 (Wang et al., 2025)[2]. We denote our structure-aligned ESM2 and AMPLIFY models as **SaESM2** and **SaAMPLIFY**.

### 4.3 Supervised Downstream Task Performance

To evaluate the effectiveness of our structure alignment, we benchmark our models against their unaligned counterparts on a comprehensive suite of supervised downstream tasks. For these tasks, the pLM is fine-tuned, either with a head and frozen backbone or with full-model fine-tuning, for each specific objective. We group these into structural property prediction, supervised mutation effect prediction, and broader functional property prediction.

Whenever possible we report either confidence intervals at 95% computed with bootstrapping for downstream property prediction tasks, or the full distributions over the 217 sub-benchmark assays of ProteinGym (see §4.4).

#### 4.3.1 Structural Property Prediction

**Tasks** To test the hypothesis that structure-aligned models capture more nuanced insights of protein structures, we evaluate on the following structure prediction tasks from xTri-moPGLM (Chen et al., 2024): (1) **Contact**: two residues are considered in contact if their $C_\alpha$ atoms lie within 8Å (Rao et al., 2019). We evaluate this task using Top L/5 precision, as Rives et al. (2021) , considering residue pairs with a sequence separation greater than 6 and a sequence length cutoff of 512. In order to compare existing models without data leakage, we select the subset of CASP16 proteins that have already been deposited in PDB and contain at least one long-range contact. We also report the original xTrimoPGLM test split (created from the trRosetta dataset), and (2) **Fold Classification (Fold)**: classify each protein sequence into one of $1,195$ fold classes (Hou et al., 2018), with accuracy as the evaluation metric. (3) **Secondary Structure (SS)**: assign each residue to one of three secondary structure types (Rao et al., 2019), using accuracy as the evaluation metric.

To assess the quality of the learned representations, *we freeze the backbone model and train a linear head* for 20 epochs using a batch size of 128. We use a learning rate of $1 \times 10^{-3}$, with betas set to $(0.9, 0.95)$ and a weight decay of 0.01 (Fournier et al., 2024). The linear head has a hidden size of 128, following the methodology of xTrimoPGLM. The linear head operates on residue embeddings for the token-level task (SS), on the mean-pooled residue embedding for the sequence-level task (Fold), and on pairwise residue embedding for the Contact task. We further visualize residue embeddings with secondary structure labels to assess structural alignment effectiveness in Appendix C.

**Analysis** As shown in Table 1, SaESM2 and SaAMPLIFY outperform their respective base models on all structure prediction tasks as well as existing alignment baselines on two out of three tasks, improving Contact P@L/5 on CASP16 by 59% for ESM2 and 15% for AMPLIFY. This is a direct validation of the way inter- and intra-protein structural knowledge is infused by our method. Due to the fact that ESM2-S was directly trained on fold classification with unfrozen backbone, it outperforms our alignment method on the corresponding task.

In Appendix D.1, we show that these conclusions still hold across model size and families for secondary structure prediction and, to a lesser extent, fold classification, providing proof that the method scales.

#### 4.3.2 Functional Property Prediction

We evaluate SaESM2 and SaAMPLIFY on a broad suite of downstream property prediction tasks (Xu et al., 2022; Dallago et al., 2021), which rely on structural information to some extent but are not direct structure prediction tasks. These include predictions of thermostability, metal ion binding, protein localization (DeepLoc), enzyme commission numbers (EC), gene ontology annotations (GO), and protein–protein interactions (HumanPPI) for tasks and evaluation pipeline from (Su et al., 2024).

---

[2]Note that S-PLM has approximately 100M additional parameters compared to ESM2 and all the other baselines.

Table 1: Results on supervised downstream tasks. We report the primary metric for each task. Values are formatted as Metric [95% Confidence Interval]. The best-performing model within each family (ESM2-based and AMPLIFY-based) is in **bold**. Models within the best Confidence Interval are in *italic*.

| Model | Contact (P@L/5 ↑) | | Fold | SS | Fitness | Stability |
|---|---|---|---|---|---|---|
| | trRosetta | CASP16 | Acc (↑) | Acc (↑) | Sp. (↑) | Sp. (↑) |
| ESM2 | 0.390 [0.380,0.400] | 0.181 [0.146,0.224] | 0.677 [0.662,0.692] | 0.845 [0.843,0.847] | 0.945 [0.937,0.951] | 0.744 [0.736,0.752] |
| ESM2-S | 0.387 [0.377,0.398] | 0.182 [0.148,0.222] | **0.764** [0.750,0.778] | 0.811 [0.809,0.813] | **0.961** [0.955,0.965] | 0.765 [0.756,0.773] |
| ISM | 0.426 [0.417,0.436] | 0.220 [0.181,0.262] | 0.598 [0.580,0.614] | 0.840 [0.838,0.842] | *0.957* [0.951,0.962] | 0.558 [0.545,0.571] |
| S-PLM | 0.403 [0.394,0.413] | 0.229 [0.190,0.273] | 0.662 [0.646,0.677] | 0.821 [0.819,0.823] | 0.947 [0.940,0.952] | 0.661 [0.651,0.672] |
| SaESM2 | **0.461** [0.450,0.471] | **0.288** [0.250,0.327] | 0.681 [0.665,0.696] | **0.865** [0.863,0.866] | *0.957* [0.951,0.962] | **0.820** [0.813,0.827] |
| AMPLIFY | 0.253 [0.245,0.262] | *0.155* [0.120,0.192] | 0.487 [0.470,0.502] | 0.811 [0.809,0.813] | *0.947* [0.941,0.953] | 0.713 [0.704,0.722] |
| SaAMPLIFY | **0.320** [0.311,0.328] | **0.169** [0.144,0.195] | **0.576** [0.557,0.593] | **0.849** [0.847,0.850] | **0.948** [0.941,0.953] | **0.747** [0.739,0.756] |

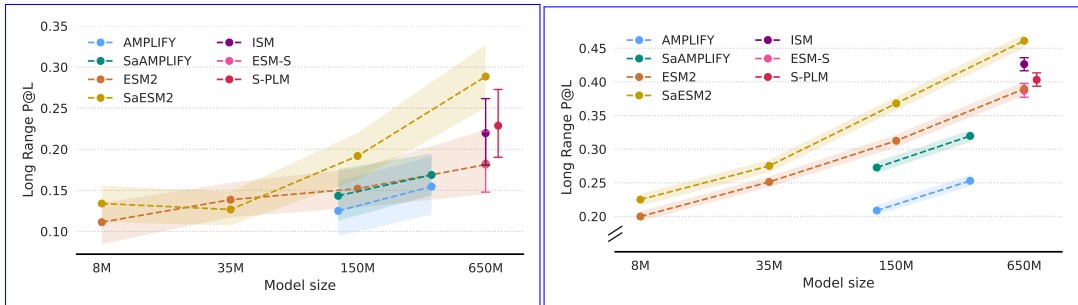

Figure 2: Longe range contact prediction precision. **(Left)** Results on the CASP16 test set. **(Right)** Results on the trRosetta test split from the xTrimoPGLM evaluation pipeline. Structure-aligned models significantly outperform their baseline on both test sets, starting at the 150M parameters model size. Note that the confidence intervals on the harder split (CASP16) are larger due to a small sample size.

In addition to downstream structural evaluations (§4.3.1) and supervised mutation effect prediction (§4.3.3) from xTrimoPGLM, we further evaluate on 3 other properties: enzyme catalytic efficiency, peptide-MHC/TCR binding affinity and peptide-HLA-MHC affinity. In all cases, we follow the data splits and training protocols from the respective papers.

As detailed in §D.1 and §D.2, we observed no meaningful change in performance between the structure-aligned models and their respective unaligned baselines, leading us to conclude that the proposed method does not degrade functional property prediction. Notably, the confidence intervals of the 3 structure-aligned models (SaESM2, ESM2-S, and ISM) fully overlap with those of their baseline across all 9 SaProt tasks. Moreover, out of the 27 points of comparison, the structure-aligned models outperform their baselines only 14 times, barely over half, suggesting that structure alignment does not provide meaningful prediction improvement or degradation on these tasks. We offer in §D.2 three possible hypotheses for this.

### 4.3.3 SUPERVISED MUTATION EFFECT PREDICTION

**Tasks** We evaluate our models on protein mutation effect prediction. Specifically, we consider two supervised tasks adopted in xTrimoPGLM: (1) **Fitness (GB1)**: predicting the binding fitness of GB1 following mutations at four specific positions; (2) **Stability**: predicting relative protease resistance as a proxy measurement for stability. For this task, evaluation is performed on one-mutation neighborhoods of the most promising proteins (Rao et al., 2019). We report performance using the Spearman correlation coefficient, while the setup is the same as in §4.3.1, except that we also fine-tune the backbone with a learning rate of $1 \times 10^{-4}$.

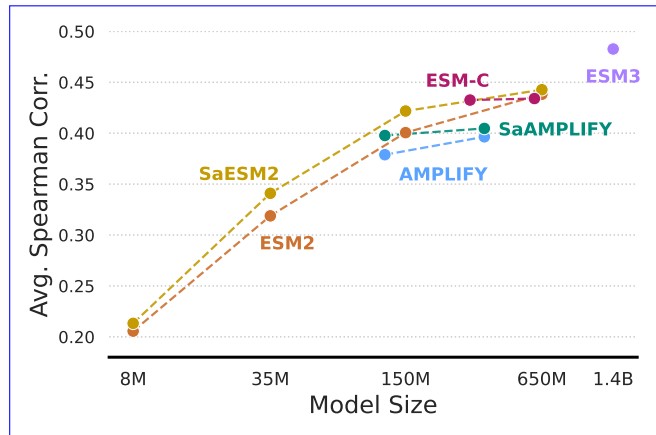

Figure 3: DMS average Spearman correlation scaling against model size for different families of models. Structure-aligned models, identified by the prefix Sa-*, consistently outperform their baseline models. SaESM2 650M and 150M are now on the Pareto front of performance against model size defined by ESM-C and ESM3.

**Analysis** As shown in Table 1, SaESM2 demonstrates a clear advantage over other models in supervised mutation effect prediction in Stability prediction [3] and a statistically competitive performance for Fitness prediction (GB1).

Additional results, across model sizes and downstream tasks can be found in Appendix D.1, providing additional insights into how the method scales. In the same section, we also report our statistically inconclusive results on Fluorescence prediction under protein mutations.

### 4.4 ZERO-SHOT DEEP MUTATIONAL SCANNING

**Tasks** We evaluate all our models on zero-shot deep mutational scanning (DMS), which is related to the supervised mutation effect prediction tasks in §4.3.3 but evaluated in a zero-shot setting. Here, we use the model's masked language modeling head to score mutations, comparing the log-likelihood of the mutated amino acid to the wild-type. We use the ProteinGym (Notin et al., 2023) DMS substitution benchmark, which compares predicted scores to experimental fitness scores for 217 assays. We used public figures for the ESM2, ESM-C and ESM3 families of models.

**Analysis** As shown in Figure 3, the structure-aligned models significantly outperform their unaligned counterparts across model families and sizes.

This consistent improvement in zero-shot fitness prediction is a strong indicator of an increase in biophysical understanding.

Protein language models rely on sequence co-evolution to predict fitness, that is, essentially predicting fitness based on "what amino acid is common at this position?". In contrast, DMS assays measure experimental fitness, which is often dominated by protein stability. Our structure-aligned models overcome this limitation. By infusing inter-protein and intra-protein structural knowledge, they constrain the MLM head to favor mutations that are not just sequentially plausible but also structurally sound. A mutation that would destabilize the protein's fold is now correctly assigned a lower probability, leading to a stronger correlation with the experimental fitness data. Additional details about ProteinGym, including violin plots for all model and a head-to-head comparison are provided in Appendix D.3.

---

[3]Note that standard deviation of the downstream evaluation pipeline for Stability prediction is very high (see Table 9 in Appendix)

## 4.5 PSEUDO-PERPLEXITY

To measure the impact of the structure alignment on the pLMs' sequence-level knowledge, we compute the pseudo-perplexity distributions of ESM2, AMPLIFY, and their structure-aligned variants as defined in Section 1.2.2 of Lin et al. (2022) using the validation set from Fournier et al. (2024). This set includes proteins with experimental evidence from reference proteomes based on high-quality genomes across all three domains of life and is designed to reflect accurately the natural protein distribution.

Figure 4 reveals that our structure alignment does increase pseudo-perplexity, indicating a trade-off in which structural integration slightly compromises sequence modeling. However, despite this, both SaESM2 650M and SaAMPLIFY 350M maintain competitive pseudo-perplexity scores, suggesting that structure alignment largely preserves the original pLMs sequence-level knowledge.

Table 2: Mean Perplexity (PPL) on the test set with 95% Confidence Intervals. T-tests compare the base model (ESM2, AMPLIFY) against its corresponding Sa-variant (SaESM2, SaAMPLIFY).

| Family | Model | Mean PPL [95% CI] | t-statistic | p-value |
|--------|-------|-------------------|-------------|---------|
| ESM2 | ESM2 (650M)
SaESM2 (650M) | 5.89 [5.81, 5.96]
6.38 [6.29, 6.46] | -8.65 | $5.50 \times 10^{-18}$ |
| AMPLIFY | AMPLIFY (350M)
SaAMPLIFY (350M) | 4.58 [4.50, 4.65]
5.10 [5.02, 5.18] | -9.30 | $1.55 \times 10^{-20}$ |

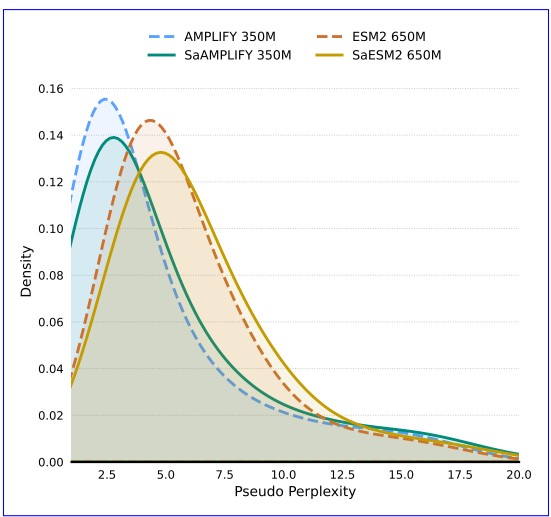

Figure 4: Pseudo-perplexity distributions on the validation set introduced by Fournier et al. (2024).

## 5 CONCLUSION

In this work, we propose to enrich sequence-only pLMs with structural knowledge. We incorporate structural insights from pre-trained pGNNs into pLMs via a latent-level task, aligning residue representations across models. To infuse intra-protein structural knowledge, we introduce a physical-level task that trains pLMs to predict structural tokens. Additionally, we propose a residue loss selection module that identifies and emphasizes challenging yet reliable residue losses to guide learning. We validate our structure alignment approach on two pLMs, ESM2 and AMPLIFY, demonstrating improved performance across diverse downstream tasks, from structure prediction to supervised and unsupervised mutation effect prediction tasks. These results suggest that structure alignment could become an indispensable component for future pLMs.

## ETHICS STATEMENT

Protein language modeling has broad applications in protein-based drug discovery. Improved downstream property prediction can enhance the accuracy and efficiency of therapeutic design–for example, more precise protein–protein affinity prediction may facilitate the development of better antibodies. However, as with many advances in biotechnology, there is a risk that such methods could be misused, including for the design of harmful agents such as biochemical weapons. As researchers, we recognize the importance of being mindful of potential misuse and weighing the benefits of our work against the risks. In the case of this paper, we believe the potential for positive impact, particularly in accelerating biomedical research and therapeutic development, far outweighs the risks. We do not identify any immediate ethical concerns associated with the research presented.

## REPRODUCIBILITY STATEMENT

Our works builds on publicly available models architectures and weights. The alignment dataset is also public and widely used for computational biology related to protein structures. We reckon the description of our method is in itself enough to reproduce the post-training phase we conduct on pLMs. Finally, all evaluation benchmarks are public. When possible, we use existing implementations of the finetuning, without further optimization for our models.

We will release post-trained model weights to HuggingFace upon acceptance of the paper, as well as open source the code. The processed dataset, including the pre-tokenization steps will also be released on HuggingFace.

## USE OF LARGE LANGUAGE MODELS

Large Language Models (LLMs) were used to aid or polish writing of this paper. They were not used for research ideation or finding related works.

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

# A    ABLATION STUDIES

We conduct extensive ablation studies on three tasks covering structure (Contact), mutation effect (Fluorescence), and property (Metal Bind) to evaluate the contribution of each design component.

**Dual-Task Framework**    Our default setup employs a weighted combination of three losses: masked language modeling, latent-level, and physical-level, with weights $(1, 0.5, 0.5)$, respectively. To assess the impact of each component, we experiment with the following configurations:

- *w/o latent*: Remove the latent-level loss, using weights $(1, 0, 0.5)$.
- *w/o physical*: Remove the physical-level loss, using weights $(1, 0.5, 0)$.
- *w/o dual*: Exclude both auxiliary losses, i.e. MLM fine-tuning on PDB, using weights $(1, 0, 0)$.

Table 3: Ablations on dual-task framework.

|  | Contact on the trRosetta split | Fluorescence | Metal Bind |
|---|---|---|---|
|  | **P@L/5** ($\uparrow$) | **Spearman** ($\uparrow$) | **Acc%** ($\uparrow$) |
| **SaESM2** (*all*) | **61.0** | **0.695** | **72.3** |
| *w/o latent* | 53.7 ($-12.0\%$) | 0.689 ($-0.9\%$) | 69.5 ($-3.8\%$) |
| *w/o physical* | 59.1 ($-3.1\%$) | 0.691 ($-0.6\%$) | 71.0 ($-1.8\%$) |
| *w/o dual* | 51.4 ($-15.7\%$) | 0.686 ($-1.3\%$) | 67.1 ($-7.2\%$) |
| ESM2 (*baseline*) | 54.1 ($-11.3\%$) | 0.687 ($-1.2\%$) | 70.8 ($-2.1\%$) |

As shown in Table 3, removing any loss term leads to performance degradation across all three tasks, confirming the effectiveness of our dual-task framework. Notably, the *w/o latent* setting performs worse than *w/o physical*, suggesting that the latent-level task contributes more significantly to the considered downstream tasks than the physical-level task. This supports our motivation that the physical-level task acts primarily as a structural constraint rather than as a dominant learning signal.

**Residue Loss Selection**    We compare our *residue-level selection* module with two alternative strategies that do not rely on reference models, instead selecting residues based solely on their individual loss values:

- *loss-large*: Select residues with high losses, assuming they offer greater learning potential.
- *loss-small*: Select residues with low losses, assuming they are cleaner and more accurate.

For comparison, we also include a *full* strategy that uses all residue losses without any selection.

As shown in Table 4, alternative selection strategies led to decreased performance across all tasks, demonstrating the effectiveness of our *residue loss selection* module. While beneficial, its impact is less significant than that of the *dual-task framework*, likely due to the already high quality of

Table 4: Ablations on residue loss selection.

| | Contact on the trRosetta split | Fluorescence | Metal Bind |
|---|---|---|---|
| | P@L/5 (↑) | Spearman (↑) | Acc% (↑) |
| **SaESM2** (*residue-loss selection*) | **61.0** | **0.695** | **72.3** |
| *loss-large* | 60.6 (−0.7%) | 0.693 (−0.3%) | 71.3 (−1.4%) |
| *loss-small* | 59.4 (−2.6%) | 0.691 (−0.6%) | 71.0 (−1.8%) |
| *full* | 60.3 (−1.1%) | 0.690 (−0.7%) | 71.1 (−1.7%) |
| ESM2 (*baseline*) | 54.1 (−11.3%) | 0.687 (−1.2%) | 70.8 (−2.1%) |

the protein structures used and the extensive pre-training of base pLMs. We further visualize the validation loss curves for different loss selection strategies in §E, which further supports the superior effectiveness of our strategy.

**Structure Embedding** We further ablate the structure embeddings used in the latent-level task. In addition to our default GearNet embeddings (Zhang et al., 2023), we explore embeddings from the AlphaFold2 Evoformer model (Jumper et al., 2021), denoted as *AF2*. Specifically, we provide the protein structure as a template and perform only one Evoformer cycle to extract the embeddings to reduce computational cost.

Table 5: Ablations on structure embedding.

| | Contact on the trRosetta split | Fluorescence | Metal Bind |
|---|---|---|---|
| | P@L/5 (↑) | Spearman (↑) | Acc% (↑) |
| **SaESM2** (*GearNet*) | **61.0** | **0.695** | **72.3** |
| *AF2* | 48.4 (−20.7%) | **0.695** (−0.0%) | 69.0 (−4.6%) |
| ESM2 (*baseline*) | 54.1 (−11.3%) | 0.687 (−1.2%) | 70.8 (−2.1%) |

As shown in Table 5, aligning with GearNet embeddings outperforms aligning with AlphaFold2 embeddings on both Contact Prediction (trRosetta split) and Metal Bind tasks. We also observed a degradation of our method when aligning to AF2 embeddings compared to the baseline ESM2 model without structural alignment. This observation is consistent with the findings of Hu et al. (2022), which suggest that embeddings from the AF2 may not be well-suited for some downstream tasks.

**Structure Token** We further ablate the structure token used in the physical-level task. Our approach is based on *foldseek* structure tokens (van Kempen et al., 2022) and we explore *protoken* (Lin et al., 2023a) and *aido* (Zhang et al., 2024a), both of which employ a larger codebook size (512 compared to 20 for *foldseek*). We do not compare against the *ESM3* structure token (Hayes et al., 2024) due to its strict commercial license.

Table 6: Ablations on structure token.

| | Contact on the trRosetta split | Fluorescence | Metal Bind |
|---|---|---|---|
| | P@L/5 (↑) | Spearman (↑) | Acc% (↑) |
| **SaESM2** (*foldseek*) | 61.0 | **0.695** | **72.3** |
| *protoken* | 60.8 (−0.3%) | **0.695** (+0.0%) | 71.9 (−0.6%) |
| *aido* | **61.9** (+1.5%) | **0.695** (+0.0%) | 70.5 (−2.5%) |
| ESM2 | 54.1 (−11.3%) | 0.687 (−1.2%) | 70.8 (−2.1%) |

As shown in Table 6, *aido* outperforms *foldseek* on the contact prediction task, likely due to its finer-grained structural representation that injects richer structural insights into the pLM. *Protoken* performs slightly worse despite its larger codebook, likely due to *protoken* encoding global dependencies instead of emphasizing local neighborhoods like *foldseek*, which aligns more closely with our structure alignment approach. This observation is consistent with that of Zhang et al. (2024a).

For the property prediction task Metal Bind, *foldseek* performs best, supporting the importance of local structure. All three tokens perform similarly on the fluorescence prediction task.

# B  RELATED WORK

## B.1  STRUCTURE LANGUAGE MODELS

There are two main types of structure language models. The first requires explicit structural input, such as structure tokens (Su et al., 2024; Heinzinger et al., 2024; Li et al., 2024) or torsion angles (Frolova et al., 2024) or geometric graphs (Hartout et al., 2025). However, these models depend on potentially unreliable or inaccurate structural data (protein structures are generally modeled at cryogenic temperatures and fail to take into account the full conformational landscape) and protein structure databases like the PDB are much smaller than sequence-only databases. Additionally, many proteins lack a well-defined, rigid structure, having disordered domains. All these approaches are not directly comparable to ours as their models requires explicit structural inputs, whereas SaESM operates purely on sequences, a property we believe is important.

The second type only requires protein sequences as input and integrates structural insights during pre-training. For example, Zhang et al. (2024b) introduces a physical-level task for fold prediction, though it is somewhat coarse. Sun & Shen (2024) proposes several physical-level tasks, including secondary structure and distance map predictions, to incorporate structural knowledge into the pLM, while Ouyang-Zhang et al. (2024) focuses on structure token prediction. Penaherrera & Koes (2024) uses a similar contrastive learning loss, but limits its focus to masked residues and does not utilize advanced pre-trained GNN models. Wang et al. (2025) and Su et al. (2025) primarily focus on latent embedding alignment and do not incorporate a physical-level task, which our ablation shows to be crucial (Table 3). Furthermore, their contrastive learning is performed at the protein-level, in contrast to our latent-level task that operates at the residue-level. We compare against S-PLM (Wang et al., 2025) as one of our baselines, but not against ProTek (Su et al., 2025) because of data leakage between their pretraining data and our downstream tasks. Note that the S-PLM post-training method adds approximately 100M parameters to ESM2, a more than 15% increase. You & Shen (2022) is limited to residue token prediction. All existing works also discard the language modeling head, which limits to some extent their applications. On the other hand, AlphaFold2 (Jumper et al., 2021) and ESMFold (Lin et al., 2023b) use sequence encoders, namely Evoformer and ESM2, followed by structure prediction modules. However, their focus is on structure prediction, and AlphaFold2 embeddings have been shown to be less effective than ESM2 embeddings for downstream tasks (Hu et al., 2022).

While recent studies have explored how to incorporate knowledge from pre-trained pLMs into pGNNs (Zheng & Li, 2024; Chen et al., 2023; Robinson et al., 2023), their focus is on improving pGNNs rather than pLMs, and no prior work has explored integrating structural insights from pre-trained pGNNs into pLMs. Our work bridges this gap by introducing the latent-level task, thereby enriching the pLMs with comprehensive structural insights from pre-trained pGNNs. Finally, the idea of distilling structural information via some form of contrastive learning between sequences is not new, with (Bepler & Berger, 2021) directly predicting contact inside a protein while simultaneously contrasting SCOP (Chandonia et al., 2017) information between protein pairs, with a language modeling trunk.

## B.2 DATA SELECTION

Data selection is a critical component in training protein models. AlphaFold2 (Jumper et al., 2021) filters proteins with a resolution higher than 9Å and excludes sequences where a single amino acid accounts for over 80% of the input sequence. Additionally, it samples protein chains based on length to rebalance distribution and cluster size to reduce redundancy, which risks deviating from the natural distribution shaped by evolutionary selection. ESM2 (Lin et al., 2023b) adopts comparable sampling strategies while AMPLIFY (Fournier et al., 2024) curates a validation set of proteins with experimental evidence at the protein or transcript level from reference proteomes derived from high-quality genomes across all three phylogenetic domains, aiming to better represent the natural protein distribution.

Data selection has also been extensively explored in natural language model pre-training, incorporating techniques such as filtering, heuristics, and domain-specific selection (Albalak et al., 2024). Our *residue loss selection* module is inspired by prior work (Lin et al., 2024), which uses excess loss to identify useful tokens in language pre-training. However, our approach differs significantly by operating at a finer granularity through residue-level loss. Given the multi-loss structure of our framework, where each residue incurs three types of losses, we focus on those with high excess loss in each specific category. Crucially, our work is rooted in the protein research rather than natural language, reflecting the unique challenges and requirements of protein modeling.

## C RESIDUE EMBEDDING VISUALIZATION

In order to qualitatively assess the effectiveness of our structure alignment technique, we visualize the residue embeddings extracted from the final layer of ESM2 and AMPLIFY before and after aligning them. Specifically, we analyze $1,000$ proteins from the Secondary Structure task, where each residue is color-coded based on its annotation to one of three secondary structure labels. We use UMAP (McInnes et al., 2018) to project high-dimensional data into a two-dimensional space with 50 nearest neighbors.

Table 7: Quantitative evaluation of embedding separability. We report the **Silhouette Score** (measure of cluster cohesion/separation between -1 and 1, higher is better) and **k-NN Classification Accuracy** ($k = 20$) for residue type, grouped residue properties as in Figure 6, and Secondary Structure (Q3) labels. **Bold** values indicate the best performance within each model family.

| Model | Silhouette Score | | | k-NN Accuracy ($k = 20$) | | |
|---|---|---|---|---|---|---|
| | Amino Acid | Grouped AA | Sec. Str. (Q3) | Amino Acid | Grouped AA | Sec. Str. (Q3) |
| ESM2 | 0.023 | 0.005 | -0.001 | **0.981** | **0.983** | 0.772 |
| SaESM2 | **0.030** | **0.010** | **0.012** | 0.964 | 0.967 | **0.861** |
| AMPLIFY | 0.053 | **0.029** | -0.007 | 0.929 | 0.946 | 0.644 |
| SaAMPLIFY | **0.058** | 0.027 | **0.009** | **0.956** | **0.964** | **0.798** |

As shown in Figure 5, applying structure alignment improves the discrimination between secondary structures. In particular, the aligned embeddings (SaESM2 and SaAMPLIFY) exhibit clearer separation compared to their unaligned counterparts. Additionally, Figure 6 shows that amino acids sharing similar physical properties are located closer in the embedding space for aligned models compared to the unaligned baseline.

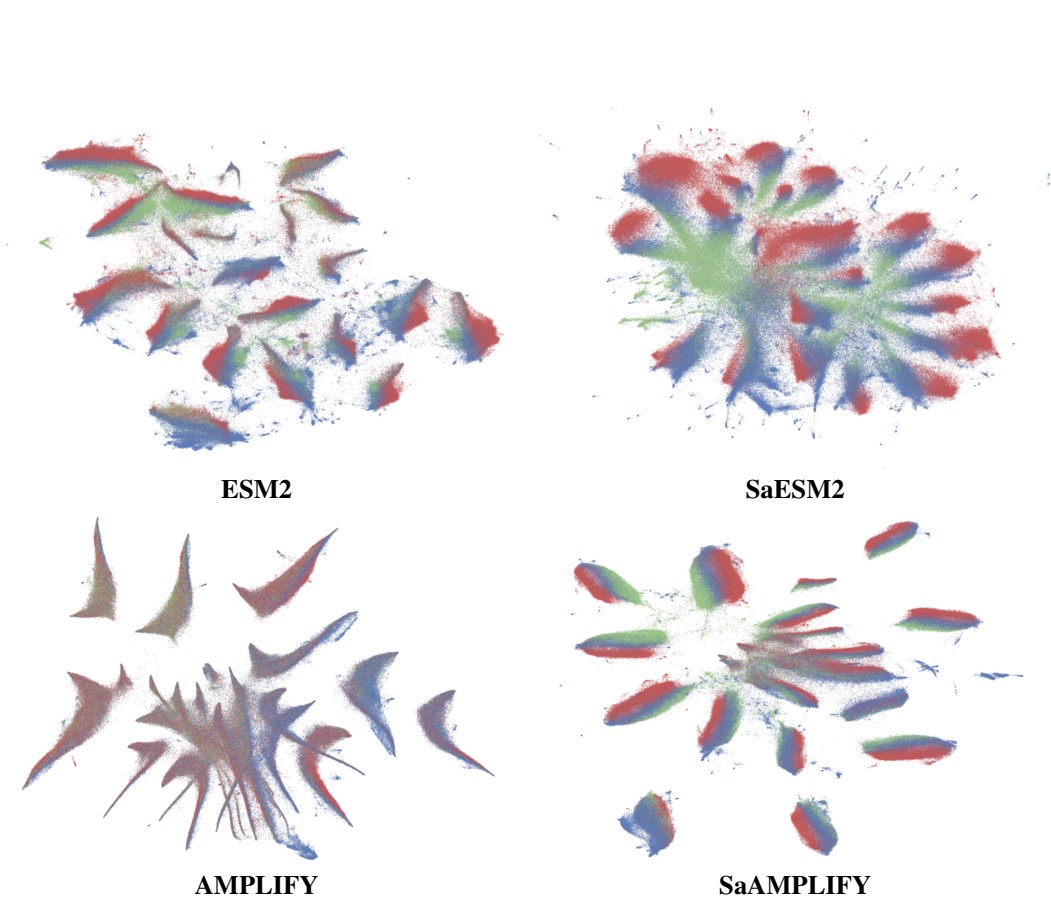

**ESM2**    **SaESM2**

**AMPLIFY**    **SaAMPLIFY**

Figure 5: Residue embeddings colored by secondary structure type colored in blue, red, and green across four models: ESM2, SaESM2, AMPLIFY and SaAMPLIFY.

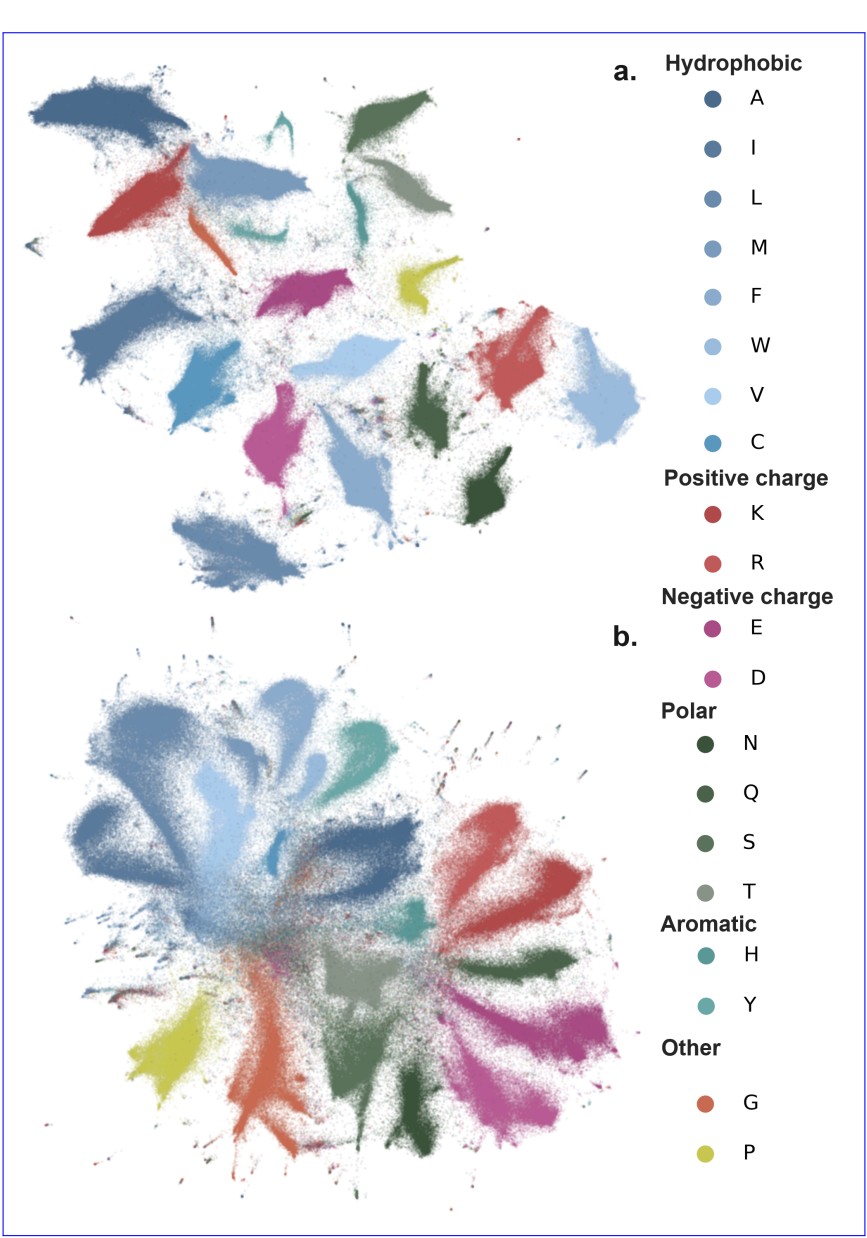

Figure 6: Residue embeddings colored by amino acid type for ESM2 (**a.**) and SaESM2 (**b.**). Amino acids with similar physical properties are colored with a gradient of the same color. Embeddings for SaESM2 clearly show a latent space more physically coherent.

# D    ADDITIONAL RESULTS AND SCALING ANALYSIS

We provide in this section additional results and visualizations on downstream property prediction benchmarks. As shown in the figures, our method scales favorably on tasks where scaling model size helps. On all tasks, considering confidence intervals, our method's performance is either comparable or higher than the base model performance. In Figure 11, we provide full violin plots for all our ProteinGym evaluations.

## D.1    FULL RESULTS ON DOWNSTREAM PROPERTY PREDICTION: XTRIMOPGLM

In this section, we report the full results, across model sizes on every evaluation from xTrimoPGLM we conducted. In Figure 7, we show that, after averaging scores on xTrimoPGLM evaluations, struture-aligned models outperform their baseline, with the largest gaps observed starting in the 100M parameters range. In Figure 8, we plot for every task all of our results.

Considering structure based tasks that are not contact map (already discussed in Figure 2), we find consistent improvements in secondary structure prediction across all model sizes. For fold prediction for all SaESM2 models up to the 150M parameters size have significantly higher accuracy. For the 650M model, confidence intervals overlap while ESM-S, for which the full backbone was trained in remote homology detection, remains best. SaAMPLIFY models always outperform their baseline.

In term of supervised mutation effect prediction, we are only able to report results for all model size for stability prediction, where SaESM2 has higher accuracy at both the 150M and 650M model sizes.

Finally, for the three last tasks: peptide-HLA MHC affinity, TCR-pMHC affinity and enzyme catalytic efficiency, we cannot report significant results.

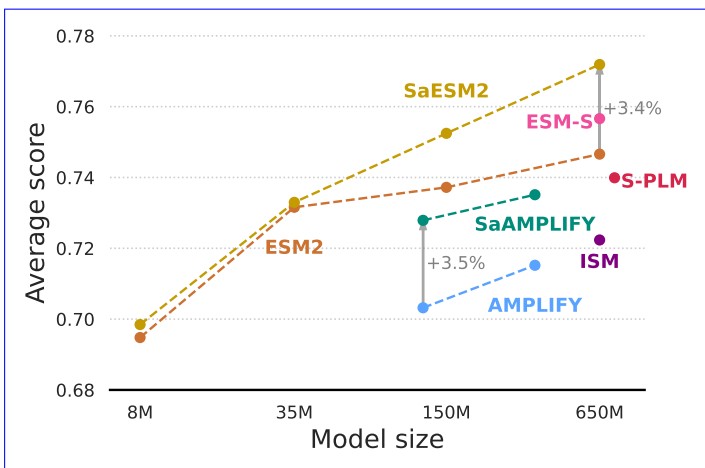

Figure 7: Average model performance compared with model size for different families of models over xTrimoPGLM. Gaps between models would widen if downstream evaluations with fully overlapping confidence intervals between all methods and baselines were removed.

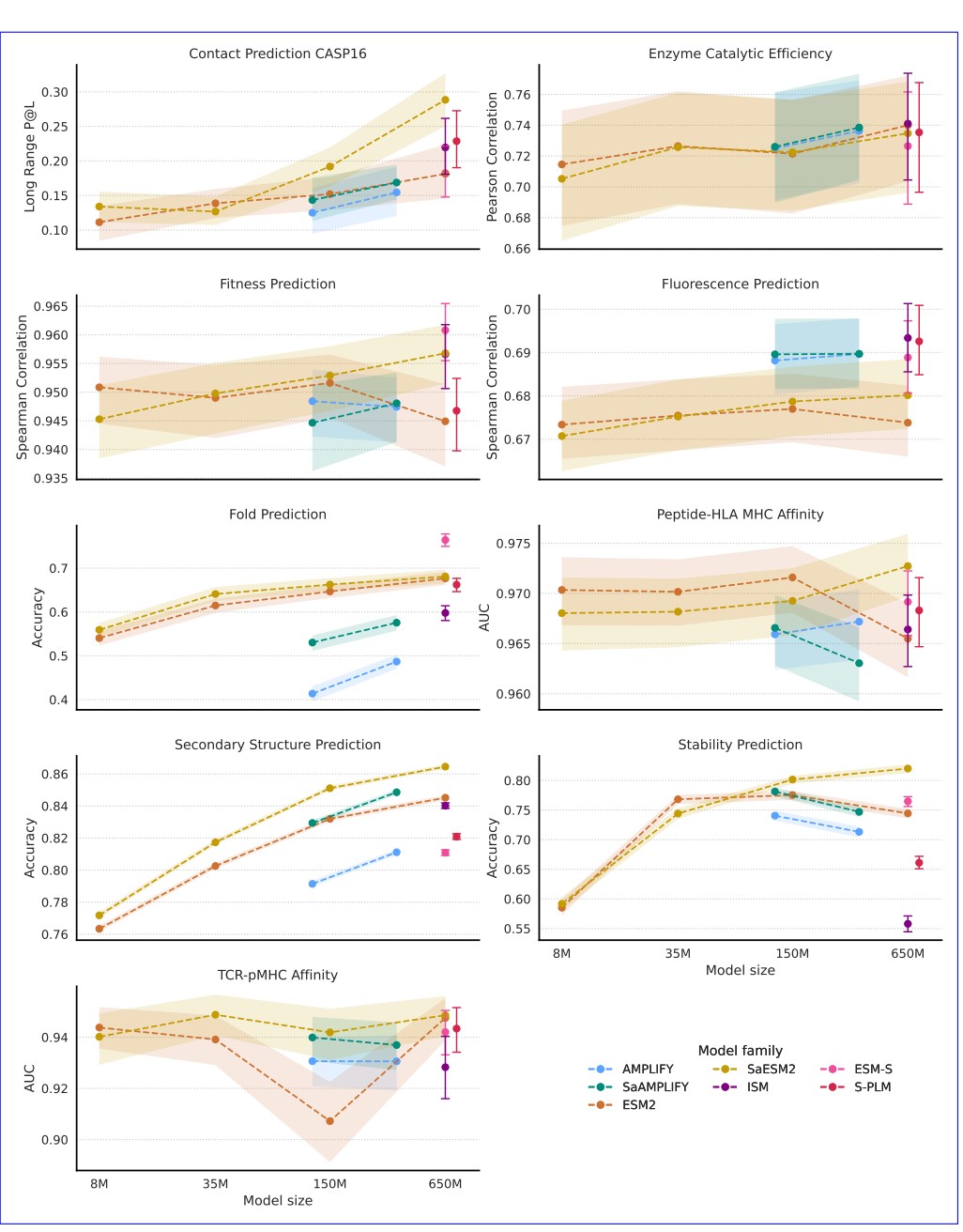

Figure 8: Model performance compared with model size for different families of models over every xTrimoPGLM task.

## D.2 FULL RESULTS ON DOWNSTREAM PROPERTY PREDICTION: SAPROT

In this section, we report the full results, across model sizes on every evaluation from SaProt we conducted. As shown in Figure 9, no improvement can be observed with structure alignment. Only a general scaling of performance with model size is visible.

The full results are displayed in Figure 10, with figures at the largest model scales in Table 8. No model outperforms significantly any of the others. For most of the downstream evaluations, confidence intervals bootstrapped from the test set are completely overlapped. We offer three possible interpretations, both for SaProt and for inconclusive xTrimoPGLM tasks: (i) these downstream tasks either do not exhibit or do not benefit from transfer learning of a foundation model (ii) the training data for these benchmarks tasks is fairly noisy (iii) the test split size for these tasks is too small compared to their noise levels, yielding large confidence intervals.

Table 8: Results on functional property prediction. Values are Metric [95% Confidence Interval]. Within each model family (ESM2-based and AMPLIFY-based), the **best-performing** model is bolded. Models in *italics* have a mean score that falls within the 95% CI of the best model in their family. SaESM2 is within confidence intervals for all tasks where it's not best, apart from GO Cellular Component and Human PPI. Confidence intervals for models (†) and tasks (†) are statistical upper bounds.

| Model | EC
Fmax (↑) | GO (BP)
Fmax (↑) | GO (CC)
Fmax (↑) | GO (MF)
Fmax (↑) | Thermostability
Sp. (↑) |
|---|---|---|---|---|---|
| ESM2 | 0.855 [0.841, 0.869] | *0.477* [0.467, 0.487] | *0.484* [0.474, 0.493] | *0.672* [0.661, 0.681] | **0.712** [0.573, 0.765] |
| ESM-S† | 0.861 [0.844, 0.878] | *0.479* [0.462, 0.496] | 0.458 [0.441, 0.475] | **0.673** [0.657, 0.689] | *0.683* [0.658, 0.708] |
| ISM† | *0.872* [0.856, 0.888] | 0.471 [0.454, 0.488] | **0.497** [0.480, 0.513] | 0.666 [0.650, 0.682] | *0.695* [0.671, 0.719] |
| S-PLM | **0.878** [0.866, 0.892] | **0.480** [0.472, 0.491] | 0.445 [0.435, 0.455] | *0.671* [0.660, 0.682] | *0.704* [0.590, 0.766] |
| SaESM2 (ours) | *0.868* [0.855, 0.882] | *0.479* [0.470, 0.489] | 0.462 [0.452, 0.473] | *0.663* [0.653, 0.674] | *0.693* [0.570, 0.756] |
| AMPLIFY | **0.501** [0.480, 0.525] | **0.271** [0.263, 0.279] | 0.322 [0.311, 0.332] | *0.378* [0.366, 0.393] | **0.614** [0.430, 0.640] |
| SaAMPLIFY (ours) | *0.486* [0.464, 0.508] | 0.257 [0.250, 0.266] | **0.348** [0.342, 0.358] | **0.389** [0.376, 0.401] | *0.596* [0.420, 0.641] |

| Model | DeepLoc (Subcell.)†
Acc (↑) | DeepLoc (Binary)†
Acc (↑) | HumanPPI†
Acc (↑) | Metal Bind†
Acc (↑) |
|---|---|---|---|---|
| ESM2 | *0.839* [0.825, 0.852] | *0.931* [0.919, 0.942] | 0.783 [0.722, 0.836] | 0.705 [0.671, 0.740] |
| ESM-S† | 0.828 [0.814, 0.842] | **0.934** [0.923, 0.945] | *0.826* [0.771, 0.875] | 0.711 [0.677, 0.745] |
| ISM† | 0.826 [0.812, 0.840] | *0.923* [0.910, 0.935] | *0.815* [0.759, 0.866] | 0.699 [0.665, 0.734] |
| S-PLM | **0.847** [0.834, 0.860] | *0.930* [0.917, 0.942] | **0.853** [0.799, 0.902] | 0.696 [0.661, 0.731] |
| SaESM2 (ours) | *0.840* [0.826, 0.853] | *0.933* [0.921, 0.944] | 0.777 [0.716, 0.831] | **0.759** [0.726, 0.790] |
| AMPLIFY | **0.689** [0.672, 0.706] | 0.861 [0.845, 0.877] | *0.690* [0.623, 0.756] | **0.621** [0.584, 0.657] |
| SaAMPLIFY (ours) | *0.674* [0.656, 0.691] | **0.881** [0.865, 0.895] | **0.734** [0.669, 0.796] | *0.601* [0.564, 0.637] |

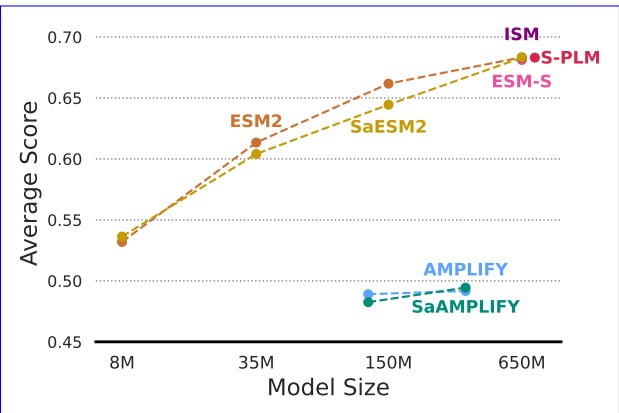

Figure 9: Average model performance compared with model size for different families of models over SaProt. The comparatively larger gap between AMPLIFY and ESM based models between the SaProt and xTrimoPGLM evaluation is probably the consequence of observed hyper-parameter sensitivity in the former evaluation pipeline.

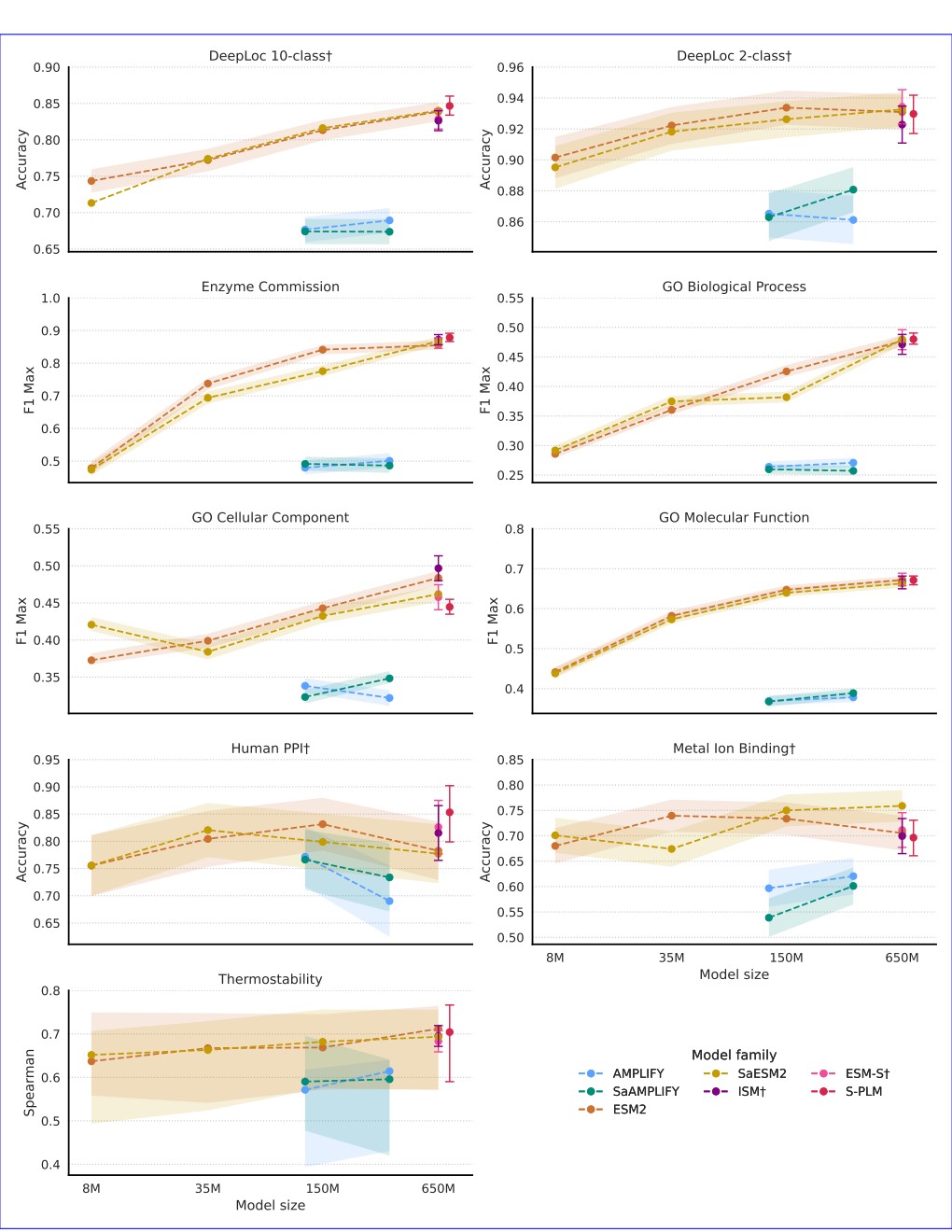

Figure 10: Model performance compared with model size for different families of models over every xTrimoPGLM task. Confidence intervals for models or tasks with † are statistical upperbounds on confidence intervals. For other tasks and models, confidence intervals are computed via bootstrapping on the test set.

## D.3 ADDITIONAL RESULTS ON PROTEINGYM

In Figure 11, we present full violin plots for all models involved in our comparison, while Figure 12 compares the performance of our models on every assay for the original model and the structure aligned one. For smaller size models, our method seems to reduce the number of assays where our fitness predictions are anti-correlated with the ground truth. For every model family and size, the improvements seem to be due in equal part to: (i) a slight improvement on average on most of the assays (ii) some assays where Spearman correlations are substantially improved. A closer inspection of these assays over the three standard ProteinGym assay metadata information (Taxon, MSA depth and Selection type) reveals no correlation between the assay type and the improvement from our method.

Overall, our aligned SaESM2 outperforms ESM2 3B and 15B (older and bigger models) and both ESM-C 300M and 600M (models of the new generation). ESM3 still remains better on ProteinGym, although he is significantly larger and fully multimodal.

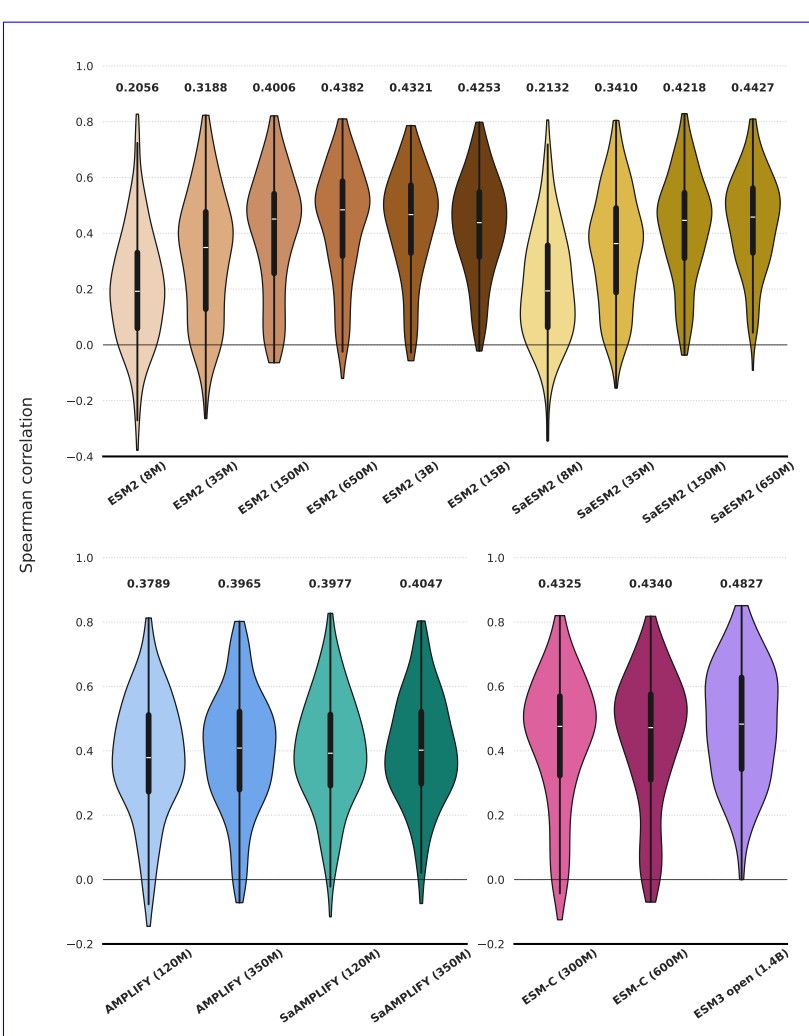

Figure 11: Violin plots of the distribution of assay spearman correlations for all models evaluated on ProteinGym. The solid black line at 0 represents the expected correlation of a random model.

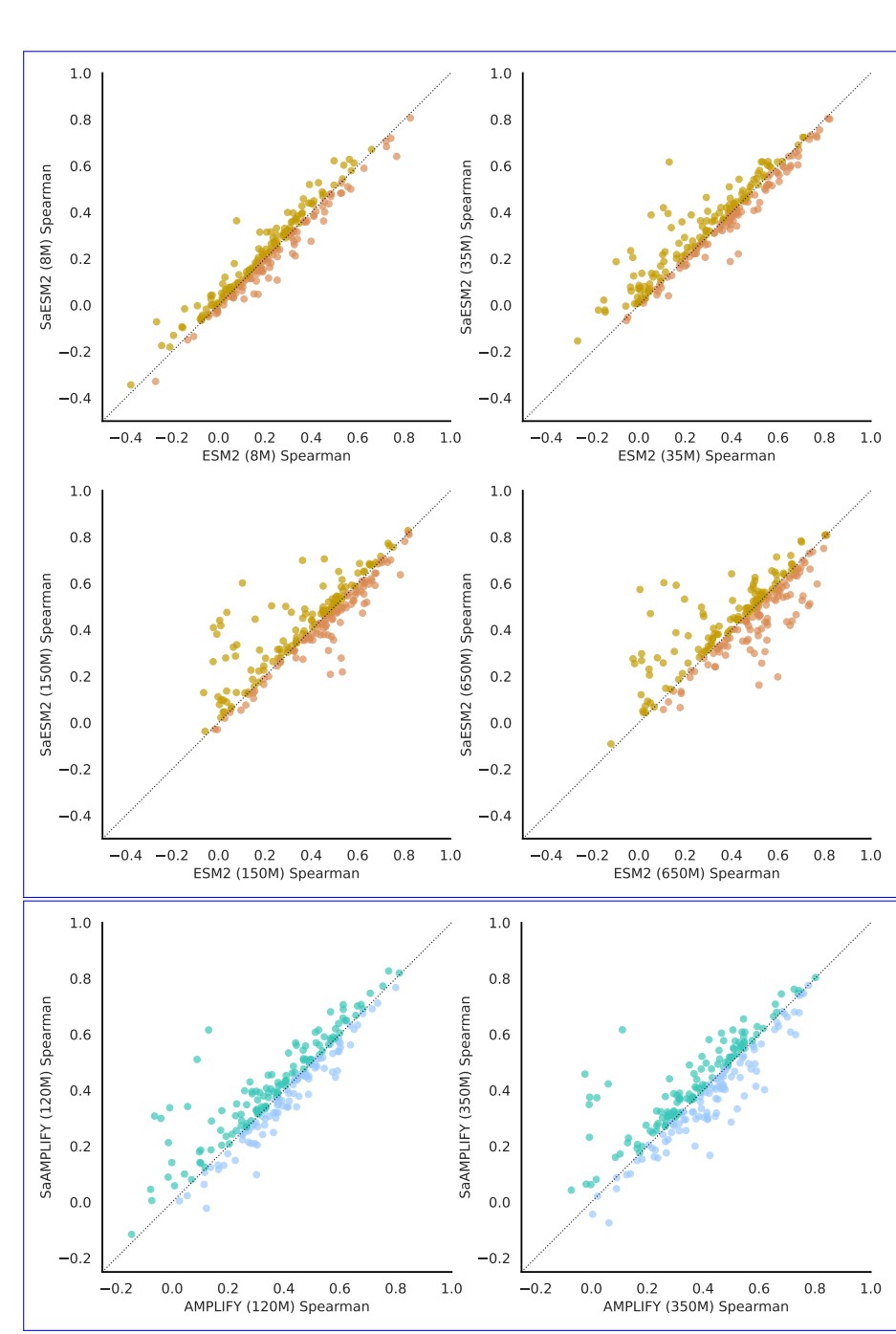

Figure 12: Head-to-head comparison of model performance on ProteinGym. Each point represents a single deep mutational scanning (DMS) assay. **(Top)** SaESM2 (y-axis) vs. ESM2 (x-axis). **(Bottom)** SaAMPLIFY (y-axis) vs. AMPLIFY (x-axis). Points above the $y = x$ diagonal (dashed line) indicate an improvement over the assay from our structure alignment.

## D.4 Stability of our post-training method: analysis of the downstream performance over 3 seeds.

Table 9: Robustness of the post-training procedure. We compare the standard deviation of the final test metric across 3 independent post-training seeds (**Std. Dev. (3 Seeds)**) against the 95% confidence interval (CI) size computed via bootstrapping on the test set (**Test Set CI Size**). The low standard deviation across seeds demonstrates the high stability of our alignment method.

| Task | Std. Dev. (3 Seeds) | Test Set CI Size |
|---|---|---|
| Contact Prediction CASP16 (Long Range P@L) | 0.0031 | 0.0204 |
| Enzyme Catalytic Efficiency (Pearson Correlation) | 0.0022 | 0.0705 |
| Fitness Prediction (Spearman Correlation) | 0.0027 | 0.0108 |
| Fluorescence Prediction (Spearman Correlation) | 0.0084 | 0.0160 |
| Fold Prediction (Accuracy) | 0.0026 | 0.0309 |
| Peptide-HLA MHC Affinity (AUC) | 0.0027 | 0.0066 |
| Secondary Structure Prediction (Accuracy) | 0.0013 | 0.0032 |
| Stability Prediction (Accuracy) | 0.0766 | 0.0185 |
| TCR-pMHC Affinity (AUC) | 0.0022 | 0.0154 |

To validate the stability of our structure-alignment procedure, we performed post-training using 3 different random seeds and measured the standard deviation of the final test metric. In Table 9, we compare this post-training variance against the inherent uncertainty of the benchmarks themselves, represented by the 95% confidence interval (CI) size derived from bootstrapping the test set. For 8 out of the 9 tasks, the standard deviation from our post-training seeds is substantially smaller than the test set CI, often by an order of magnitude (e.g., 0.0027 vs. 0.0108 for Fitness Prediction). This result is crucial, as it indicates that our method is highly robust and that the observed variance in benchmark scores is dominated by the test set's composition, not by the stochasticity of our alignment process. We note that Stability Prediction is an exception, showing higher seed variance than test set uncertainty. After further investigation, this sensitivity is mainly due to the fine-tuning pipeline, as we found our results on Stability Prediction to be particularly sensitive to hyper-parameter choices.

## E Loss Curve Analysis of Residue Loss Selection

To assess the effectiveness of our proposed *residue loss selection* module, we analyze validation loss curves across four strategies: ours, loss large, loss small, and full. These are shown in Figure 13 (overall loss), Figure 14 (MLM loss), Figure 15 (latent-level loss), and Figure 16 (physical-level loss). Recall that the overall loss is defined as:

$$\mathcal{L}_{\text{overall}} = \mathcal{L}_{\text{mlm}} + 0.5\mathcal{L}_{\text{latent}} + 0.5\mathcal{L}_{\text{physical}}. \tag{12}$$

As seen in Figure 13, our strategy consistently achieves the lowest overall loss, demonstrating superior training effectiveness and efficiency. Figure 15 shows that the primary reduction comes from the latent-level loss, indicating that our method successfully identifies informative and challenging latent-level residue losses to enhance learning. In contrast, Figure 16 shows negligible differences in physical-level loss across most strategies, except for loss small. We attribute this to the limited Foldseek codebook size (20), which provides only coarse structural information, reducing the potential benefit of residue loss selection at this level. Notably, the loss small strategy results in high physical-level loss, likely due to its focus on easy-to-learn residues, which fail to contribute meaningful structural insights to the pLMs. We further experiment with joint training on both the training and validation sets. However, this led to degraded downstream performance, likely due to overfitting on the validation set.

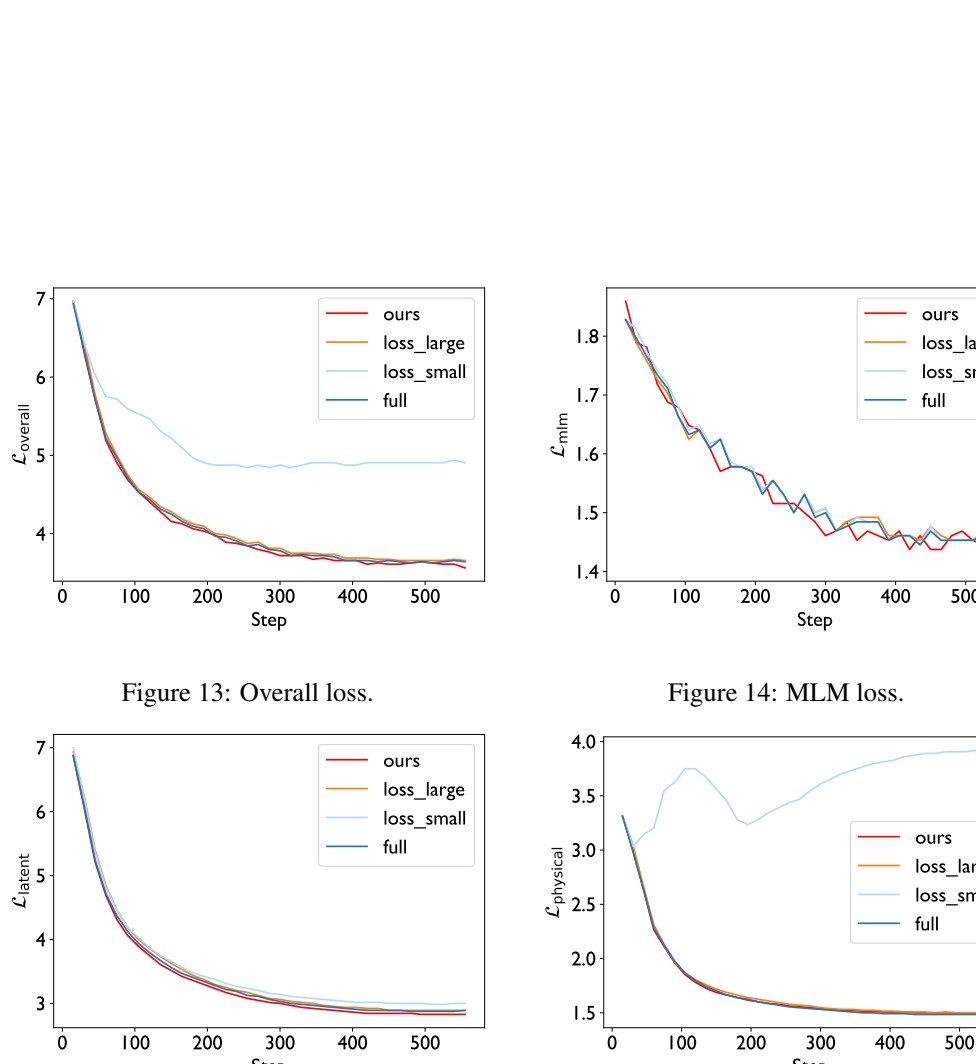

Figure 13: Overall loss.

Figure 14: MLM loss.

Figure 15: Latent-level loss.

Figure 16: Physical-level loss.

