# OpenReview forum: "Structure-Aligned Protein Language Model"
_ICLR.cc/2026/Conference — Submitted to ICLR 2026_

### Official Review · Reviewer_ZEm1 · 2025-10-30

**Soundness:** 3
**Presentation:** 3
**Contribution:** 2
**Rating:** 6
**Confidence:** 4

**Summary:**

This paper addresses a well-known limitation of large-scale protein language models (pLMs): while they excel at sequence-based tasks, they lack explicit knowledge of 3D protein structure, which is critical for many biological applications.

The authors propose a dual-task framework to finetune sequence-only pLMs with structural information. The two tasks are:
* Latent-level task: A contrastive learning objective that aligns the latent residue representations from the pLM with corresponding representations from a pre-trained, frozen protein graph neural network (pGNN), specifically GearNet. This is designed to distill *inter-protein* (dataset-level) structural knowledge into the pLM.
* Physical-level task: A prediction task where the pLM's residue embeddings are used to predict discrete structural tokens from Foldseek, reinforcing *intra-protein* (local) structural context.

In addition, the authors also propose a *residue loss selection* module mitigate the impact of low-quality structures in the PDB: they train a smaller reference model on a high-quality subset and then prioritize training the main pLM on residues with a high "excess loss" (i.e., residues that are both reliable/learnable and challenging for the current model).

The authors apply this framework to ESM2 and AMPLIFY, creating "SaESM2" and "SaAMPLIFY." Performance improvements are observed across a wide range of downstream tasks, including structure prediction, mutation effect prediction, and property prediction.

I think the paper is of high quality in terms of conceptualization, experiment design and presentation, but there are certain concerns that need to be addressed.

**Strengths:**

* The ablation studies are comprehensive. Comparisons are made with different base models, ablated loss components, ablated residue loss selection module, different structure latents and different structure tokens.
* The performance is strong, especially for the contact prediction task.
* The paper is written in an easy-to-understand language. I believe most readers of ICLR can follow the paper easily.

**Weaknesses:**

* The method is a bit intuitive and is potentially in lack of novelty. Using contrastive losses to enhance representation learning is a standard technique today. The residue loss selection module has certain novelty, but the performance improvement caused by it is marginal (comparing "SaESM2" with "full" in Table 5).
* Some terminologies could be misleading, such as "inter-protein" and "intra-protein".

**Questions:**

* How do you compare your method with SaProt, in which the structural information is directly modeled in the vocabulary?
* Have you tried using the learned protein representations in more downstream tasks, such as protein-ligand binding prediction?

---

> ### Author Response · Authors · 2025-11-21
>
> We sincerely appreciate the detailed feedback and constructive suggestions from the reviewers. We are pleased that they found the paper well-written and the methodology clear (3vm7, N8FK, ZEm1). Reviewers also highlighted that our work addresses an important problem with high application potential (3vm7, Bdnm) using a novel approach to the contrastive loss (3vm7, Bdnm) and thorough ablations (Bdnm, ZEm1). We have thoroughly considered all comments and concerns and have provided detailed responses below. Most notably, we have incorporated a scaling analysis on ESM2, from 8M to 650M, computed confidence intervals for all reported benchmarks, and added new benchmarks, including ProteinGym and CASP16, showcasing the strengths of our method.
>
> ---
>
> **Comment.** The method is a bit intuitive and is potentially in lack of novelty. Using contrastive losses to enhance representation learning is a standard technique today. The residue loss selection module has certain novelty, but the performance improvement caused by it is marginal (comparing "SaESM2" with "full" in Table 5).
>
> **Answer.** While contrastive objectives are indeed widely used today, our dual-task framework goes substantially beyond standard contrastive representation learning. The latent-level task in our design plays a role analogous to contrastive learning, but critically operates at the residue level, whereas prior work predominantly focuses on protein-level objectives and therefore fails to capture fine-grained residue-level structural knowledge. Moreover, in addition to this latent-level task that injects inter-protein residue-level structural knowledge into the pLM, we introduce a physical-level task that incorporates intra-protein residue-level structural constraints.
>
> To the best of our knowledge, our dual-task formulation is the first to jointly integrate both inter-protein and intra-protein residue-level structural information within a unified framework. Existing approaches typically capture only one dimension, either latent-level or physical-level supervision, and mainly operate at the protein level, missing the finer-grained residue-level structural knowledge. Our ablation studies further confirm that each component contributes to the final performance, underscoring the necessity of the full dual-task framework.
>
> **Comment.** Some terminologies could be misleading, such as "inter-protein" and "intra-protein".
>
> **Answer.** We agree that the terminology is not ideal. In an effort to accommodate our dual audience and after discussing with biologists and machine learning scientists, we ultimately chose "inter-protein" and "intra-protein”. We have included a footnote to clarify this terminology and its difference from common biological usage. If you have suggestions on how to improve this footnote, we would be happy to revise it.
>
> **Comment.** How do you compare your method with SaProt, in which the structural information is directly modeled in the vocabulary?
>
> **Answer.** Our approach assumes that pLMs only have access to sequences due to the real-world constraint that many proteins lack structures (e.g., about 50% of human proteins contain intrinsically disordered regions). SaProt requires additional structure tokens in input, and while it supports a wildcard as structure tokens, the authors mention that SaProt’s performance in this scenario is comparable to that of similarly sized ESM2 models (Appendix E.3 and Fig. 6 of their paper).
>
> **Comment.** Have you tried using the learned protein representations in more downstream tasks, such as protein-ligand binding prediction?
>
> **Answer.** We incorporated two peptide/MHC affinity prediction tasks from the xTrimoPGLM evaluation suite (Figure 8). Across all model scales, we did not observe statistically significant improvements, except for a few individual data points. This may be due to our post-training method’s focus on single-chain, protein-level tasks. We plan to explore methods tailored for interactions and broader downstream applications, such as protein-ligand binding and molecule-level tasks, in future research.

---

> > ### Comment · Reviewer_ZEm1 · 2025-11-27
> >
> > Thank you for the clarification. Your comments solved most of my concerns, and I would like to increase my rating.

---

> > > ### Author Response · Authors · 2025-11-27
> > >
> > > Thank you for your supportive update. We’re glad the clarifications addressed your concerns, and we sincerely appreciate your willingness to improve the rating.

---

### Official Review · Reviewer_N8FK · 2025-10-31

**Soundness:** 2
**Presentation:** 2
**Contribution:** 1
**Rating:** 2
**Confidence:** 4

**Summary:**

This paper proposes a contrastive learning-based framework to align protein language models with structural data. They introduce a protein-level and a residue-level CLIP-style post training task. To measure the improvement in 6 structure-related tasks, 2 mutation effect prediction tasks, and 9 property prediction benchmarks; they observe consistent improvements in those tasks.

**Strengths:**

Most claims are well-supported, particularly when it comes to the performance on the many downstream tasks which is consistent, albeit marginal. Overall the paper is easy to understand - the method is presented in a straightforward way,

**Weaknesses:**

Because of this marginal gain observed in the downstream tasks, subsamples could be used to obtain variances of the metric estimates to show that those increases are still substantial, and ideally statistically significant. Additionally, stating that the embeddings show better separation as a result of structural alignment based on UMAP projections need to be supplemented with more quantitative methods measuring separability. Also, Figure 2 needs to have some kind of statistical test to show the increase in pseudo-perplexity. There seem to indeed be some increase visually but this needs to be quantified more concretely.

Presentation: Major missing work includes: BioCLIP https://arxiv.org/abs/2311.18803, S-PLM https://advanced.onlinelibrary.wiley.com/doi/10.1002/advs.202404212. Similar ideas are discussed here: https://openreview.net/pdf?id=xDcTugulVV.

In terms of presentation: the embeddings presented are not very concrete and waste considerable space. Comparatively, the other parts of Figure 1 are too small and hard to read.

A non-exhaustive collection of language improvements:

- The footnote on page 1 is confusing - one could consider rephrasing to avoid biological connotations entirely.
- "structural insights" is a vague term which could be made more precise; one could consdier "embeddings derived from pGNNs"
- Abstract: "including a 12.7% increase" lacks a metric name in which this increase is observed.
- "the common types" in 2.1 is a vague phrase. Consider making this more explicit
- 4.6 "does increases"
- In the introduction, some sentences need to be broken up to improve flow. The paper overall is hard to read in its current state.
- Ensure all acronyms are explicitly defined on first use.
- There is inconsistent capitalization of e.g. protein language models.
- R-free needs to be defined, either in a footnote or supplementary material,

The paper makes overall very incremental contributions to the field - the idea of using contrastive learning on proteins is not new. More generally, I wonder if considering the structure and the sequence of the protein as different 'views' of the same underlying manifold is warranted, given one can be predicted from the other as per Anfinsen's principle (and as shown by AlphaFold, ESMFold, SimpleFold, etc.). Additionally, making pLMs "structure-aware" is known to bring advantages but mostly at a small parameter count, as larger models inherently are able to capture underlying structural features. This confines the benefits of structural alignment for pLMs primarily to models at the lower end of the size spectrum.

**Questions:**

- Given ESM-2 has many model sizes, can we see the effect of structure alignment on multiple model sizes and see if the benefits of structural alignments mostly benefit smaller models?
- Does the pGNN benefit from being post-trained in this way?
- Have the authors tried other pGNNs than GearNet?
- Can the authors clarify if the compute costs associated to performing structure alignment outperform finetuning of a base pLM on a given task with the same compute budget?
- Can the authors comment on the effects of structure alignment on larger ESM-2 model sizes like the 3B and 15B models? I would expect those effects to be more marginal.

---

> ### Author Response · Authors · 2025-11-21
>
> We sincerely appreciate the detailed feedback and constructive suggestions from the reviewers. We are pleased that they found the paper well-written and the methodology clear (3vm7, N8FK, ZEm1). Reviewers also highlighted that our work addresses an important problem with high application potential (3vm7, Bdnm) using a novel approach to the contrastive loss (3vm7, Bdnm) and thorough ablations (Bdnm, ZEm1). We have thoroughly considered all comments and concerns and have provided detailed responses below. Most notably, we have incorporated a scaling analysis on ESM2, from 8M to 650M, computed confidence intervals for all reported benchmarks, and added new benchmarks, including ProteinGym and CASP16, showcasing the strengths of our method.
>
> ---
>
> **Comment.** Because of this marginal gain observed in the downstream tasks, subsamples could be used to obtain variances of the metric estimates to show that those increases are still substantial, and ideally statistically significant.
>
> We computed bootstrapped 95% confidence intervals across all downstream tasks, which revealed that for a few tasks, particularly those from SaProt, the ESM2 baseline and all structure-aligned ESMs have overlapping confidence intervals. We discuss the reasons for this in our revised Appendix C.2. Results for the other benchmarks are statistically significant and for some substantial, including +59% improvements in P@L on CASP16, +11% on stability prediction for SaESM2 650M, and +7% and +6% on ProteinGym for the 35M and 150M models, respectively. We would like to bring the reviewer’s attention to the added CASP16 contact map prediction task, which uses data that is fully deduplicated and time-filtered, making it truly out-of-distribution and is considered a gold standard benchmark by biologists. Finally, we also added the ProteinGym benchmark, which covers 2 million mutations over 217 assays, with improvements across model sizes and families.
>
> **Comment.** Additionally, stating that the embeddings show better separation as a result of structural alignment based on UMAP projections need to be supplemented with more quantitative methods measuring separability. [...] Also, Figure 2 needs to have some kind of statistical test to show the increase in pseudo-perplexity. There seem to indeed be some increase visually but this needs to be quantified more concretely.
>
> For both these remarks, we are re-computing the results and will update the reviewer with additional metrics. We suggest computing the Mean PPL coupled with a t-test between the base and structure-aligned model. For the embeddings, we plan on computing both kNN and clustering accuracies to measure homogeneity of the residue embedding space. We would compute this accuracy both in terms of residue types but also by amino acid groups (hydrophobic, charged, polar, aromatic and others, as in Fig. 6 of the Appendix). We will also compute the silhouette score of clusters.

---

> ### Author Response · Authors · 2025-11-21
>
> **Comment.** Major missing work includes: BioCLIP [1], S-PLM [2]. Similar ideas are discussed in Multi-modal contrastive learning for proteins by combining domain-informed views [3].
>
> Thank you for highlighting [1] [2] and [3]. We would like to first clarify how our method differs from them:
> * BioCLIP. This approach fundamentally differs from the proposed method as it focuses on the vision domain. Reviewer Bdnm pointed out another paper named BioCLIP (Contrasting Sequence with Structure: Pre-training Graph Representations with PLMs [4]) that is already mentioned in our related work section. This method differs from the proposed method as it focuses on improving pGNNs, not pLMs. Notably, they maintain the pLM frozen while optimizing the pGNN, whereas our method aims to improve the pLM by injecting structural information from a frozen pre-trained pGNN. As such, evaluating them would amount to evaluating the base pLM. Furthermore, we retain the LM head and MLM task, enabling zero-shot DMS scoring. As a result, this method is not directly comparable to ours.
>
> * S-PLM. This method only considers latent embedding alignment and does not incorporate a physical-level task, which our ablations show to be important (Table 2). Furthermore, their contrastive learning operates at the protein level, while our latent-level task is conducted at the residue-level, enabling finer-grained structural supervision. They also do not make use of a pre-trained protein GNN to extract structural features, and S-PLM considers only contact maps and not structures. In comparison, our method is cheaper and produces finer-grained embeddings, which allow for a richer set of downstream tasks. Both these methods also discard the LM head, preventing any downstream language modelling tasks, such as ProteinGym.
>
> * Multi-modal contrastive learning for proteins by combining domain-informed views [3]. This method also operates at the protein level. Their training task can be divided into 3 parts: (i) contrastive learning on sequences only, harnessing positive and negatives using InterPro; (ii) contrastive learning on structures only, with data augmentations, similar to some practices in SSL for computer vision; (iii) contrastive learning between sequences and structures. They use pre-trained pGNN (GearNet) and pLMs (ESM-1b). They also discard the LM head. Code or weights for [3] is unavailable which makes it impossible for us to evaluate it on all our sequence level tasks. We can only compare tables of results they made available for the following tasks: GO - MF / BP and CC prediction and EC prediction. Reporting for [3] the best performance from Table 1 (meaning the optimal model differs from one line to the next for them).
>
> | Model | GO-BP (F1 max) | GO-MF (F1 max) | GO-CC (F1 max) | EC (F1 max) |
> | :--- | :--- | :--- | :--- | :--- |
> | **ESM-Gearnet from [3]** | 0.503 | 0.643 | 0.450 | 0.866 |
> | **SaESM2** | 0.479 [0.470, 0.489] | 0.663 [0.653, 0.674] | 0.462 [0.452, 0.473] | 0.868 [0.855, 0.882] |
>
>
> The reviewer should note that GO and EC prediction tasks were found to have high variance compared to the final reported performance for all our baselines.
>
> We indeed missed S-PLM and are currently evaluating it on the relevant benchmarks and will update you as soon as we have the results.
>
> Additionally, we propose to add the following discussion regarding S-PLM to the Related Work (see our answer to reviewer 3vm7): “S-PLM and ProTrek primarily focus on latent embedding alignment and do not incorporate a physical-level task, which our ablation shows to be crucial (Table 2). Furthermore, their contrastive learning is performed at the protein-level, in contrast to our latent-level task that operates at the residue-level.”

---

> ### Author Response · Authors · 2025-11-21
>
> **Comment.** The paper makes overall very incremental contributions to the field - the idea of using contrastive learning on proteins is not new.
>
> **Answer.** While contrastive objectives are indeed widely used today, our dual-task framework goes substantially beyond standard contrastive representation learning. The latent-level task in our design plays a role analogous to contrastive learning, but critically operates at the residue level, whereas prior work predominantly focuses on protein-level objectives and therefore fails to capture fine-grained residue-level structural knowledge. Moreover, in addition to this latent-level task that injects inter-protein residue-level structural knowledge into the pLM, we introduce a physical-level task that incorporates intra-protein residue-level structural constraints.
>
> To the best of our knowledge, our dual-task formulation is the first to jointly integrate both inter-protein and intra-protein residue-level structural information within a unified framework. Existing approaches typically capture only one dimension, either latent-level or physical-level supervision, and mainly operate at the protein level, missing the finer-grained residue-level structural knowledge. Our ablation studies further confirm that each component contributes to the final performance, underscoring the necessity of the full dual-task framework. We are also the first to keep the Language Modeling head and the MLM task. This additional regularization preserves the ability of the model to perform language modeling, for example to do zero shot mutation scoring.
>
> **Comment.** “Additionally, making pLMs "structure-aware" is known to bring advantages but mostly at a small parameter count, as larger models inherently are able to capture underlying structural features. This confines the benefits of structural alignment for pLMs primarily to models at the lower end of the size spectrum.”
> “Given ESM-2 has many model sizes, can we see the effect of structure alignment on multiple model sizes and see if the benefits of structural alignments mostly benefit smaller models?” “Can the authors comment on the effects of structure alignment on larger ESM-2 model sizes like the 3B and 15B models? I would expect those effects to be more marginal?”
>
> We have added a study on scaling our method with model size, going from ESM2-8M to ESM2-650M and from AMPLIFY 120M to 350M, adding 4 new structure aligned models (and the corresponding baselines).
> Overall, the benefits seem to materialize more in the 35M to 650M range of parameters for downstream property prédictions, with positive scaling behaviour (Fig.7, mainly driven by increase of performance on structural prediction tasks) and for Deep Mutation Scoring on ProteinGym (Fig. 3 added evaluation).
>
> Looking at the official leaderboard of ProteinGym, ESM2 650M has an average performance of 0.414, 3B of 0.406, and 15B of 0.4. This saturation also occurs for xTrimoPGLM, between the 3B and 10B models. This is also observed in “Understanding Language Model Scaling on Protein Fitness Prediction”. The paper “Medium-sized protein language models perform well at transfer learning on realistic datasets” mentions in their abstract that “While larger models, such as [ESM2 15B], promise to capture more complex patterns in sequence space, they also present practical challenges due to their high dimensionality and high computational cost. [...] We found that larger models do not necessarily outperform smaller ones, in particular when data is limited. Medium-sized models, such as ESM-2 650M and ESM C 600M, demonstrated consistently good performance, falling only slightly behind their larger counterparts—ESM-2 15B and ESM C 6B[...].” Finally, Appendix F of the paper “Diverse Genomic Embedding Benchmark For Functional Evaluation Across The Tree Of Life” also shows that ESM2-650M is often better than the 15B.
>
> As a consequence, we chose not to study ESM2 3B and 15B. They are also more expensive in terms of compute. We nevertheless expect performance to keep scaling on larger models that are not saturated.

---

> ### Author Response · Authors · 2025-11-21
>
> **Comment.** More generally, I wonder if considering the structure and the sequence of the protein as different 'views' of the same underlying manifold is warranted, given one can be predicted from the other as per Anfinsen's principle (and as shown by AlphaFold, ESMFold, SimpleFold, etc.).
>
> Thank you for the comment. While sequence and structure are theoretically linked, this relationship does not make structural information redundant for representation learning. For many downstream tasks—such as binding and stability prediction, function depends much more directly on 3D geometry than on the raw sequence. A sequence-only LM must implicitly infer such geometric regularities from 1D data, which can be inefficient (hard to learn) or incomplete. By explicitly injecting structural information, contrastive methods provide a complementary view that helps the model capture residue-level geometry more reliably, leading to improved downstream performance.
>
> **Comment.** Does the pGNN benefit from being post-trained in this way?
>
> **Answer** No. While previous works focused on pGNNs, we decided to focus on pLM as they are more general (larger databases, 50% of the proteome mass being disordered so having no rigid structure). As a result, we keep the pGNN frozen and only align the pLM. This has the benefit of making our method cheaper, as structure embeddings are only pre-computed once.
>
>
> **Comment.** Have the authors tried other pGNNs than GearNet?
>
> **Answer.** Yes. The ablation in Section 4.6 evaluates the impact of different pGNNs for embedding extraction (AlphaFold2 and GearNet) and models for generating structure tokens (Foldseek, Protoken, and Aido).
>
> **Comment.** Can the authors clarify if the compute costs associated to performing structure alignment outperform finetuning of a base pLM on a given task with the same compute budget?
>
> **Answer** Our method is designed to be a cheap post-training step that one can apply on any pLM. It is separate from fine tuning it on a given task. The goal is to improve performance across the board for as many downstream tasks as possible. The end user is still supposed to finetune the model on their task. However, our method remains cheaper than the compute spent on evaluations. From our own experience on evaluations we ran for this paper with two different pipelines (xTrimoPGLM and SaProt), it seems that the limiting factor for downstream performance of pLMs in their current form is the downstream data quantity and quality and the expressivity and quality of learned pLM representations, not the finetuning compute.

---

> ### Comment · Reviewer_N8FK · 2025-11-21
>
> Thank you for addressing my concerns, the paper is substantially stronger now than it was at the time of submission. I am waiting to view the fully updated results to further improve my score
> Meanwhile, upon examining the rebuttal version of the paper, I found missing section headers for sections 4.4 and 4.6.

---

> > ### Author Response · Authors · 2025-11-21
> >
> > Thank you for this initial increase, we remain comitted to providing you the results as soon as we have them.
> >
> > The 4.4 and 4.6 sections headers are unfortunate artefacts of using latexdiff to highlight added content. We moved and merged certain subsections and these are just the remaining section headers. The updated paper is complete and we will remove these artefacts for the camera ready version.

---

> > > ### Author Response · Authors · 2025-12-03
> > >
> > > **Comparison with S-PLM**
> > > We have finalized our evaluation of S-PLM on the different relevant benchmarks and updated Section 4 and Appendix D with the results. On average over xTrimoPGLM benchmarks, our method significantly outperforms S-PLM (+4.3%), with the most notable improvements for contact map prediction on CASP16, secondary structure prediction (5.4%) and stability prediction (+23%). Our conclusion on saturation of performance for the SaProt benchmark remains largely unchanged, with the exception of Human PPI prediction, where S-PLM slightly outperforms our model and the base ESM2 model, while being comparable with the other structure aligned baselines
> > >
> > > **Embedding space**
> > > We supplemented our UMAP visualizations with Silhouette Scores and k-NN classification accuracy ($k=20$) across residue types, physicochemical properties, and secondary structure (Table ...)
> > > The results are nuanced: while our structure-aligned models (SaESM2, SaAMPLIFY) achieve higher or comparable Silhouette Scores, confirming tighter clustering of structural features, the classification performance varies. We observe substantial gains in Secondary Structure accuracy (e.g., SaESM2 improves to 86.1% from 77.2%), but this comes with a slight trade-off in Amino Acid identity retention for SaESM2, where k-NN accuracy drops marginally (98.1% to 96.4%). This suggests that while structural injection enhances geometric separability, it is not uniform in the embedding space.
> > >
> > > **Pseudo Perplexity**
> > > We have added average pseudo perplexity with confidence intervals and t-tests in Table 2, Section 4.5.

---

### Official Review · Reviewer_Bdnm · 2025-10-31

**Soundness:** 3
**Presentation:** 3
**Contribution:** 3
**Rating:** 6
**Confidence:** 4

**Summary:**

The authors retrain a protein language model (PLM) so that its embeddings receive structural signals. To do this, they train a PLM such as ESM and a frozen structure-based protein embedding model (pGNN) on a few different objectives. These include aligning the sequence and structure-based embeddings, predicting structure tokens from tokenizers such as 3Di, and a specialized loss for low resolution residues. The embedding model is tested on various tasks including mutation effect prediction, contact prediction, and various protein property prediction tasks with solid results. There is also an interesting analysis on the tradeoff between perplexity and the degree of influence from structural information.

**Strengths:**

* Model architecture is well-motivated, the resolution-based loss is novel and ablations are thorough.
* Improved PLMs have very high application potential.

**Weaknesses:**

* There is a related work which could bear including in the baseline [1]
* The authors should explore a direct feature fusion approach. It is not clear what is left once even the dual loss is removed and is probably not the strongest form of this ablation. [2]


1. Chen, Dexiong, et al. "Endowing protein language models with structural knowledge." arXiv preprint arXiv:2401.14819 (2024).
2. Dai, Yimian, et al. "Attentional feature fusion." Proceedings of the IEEE/CVF winter conference on applications of computer vision. 2021.

**Questions:**

see weaknesses

---

> ### Author Response · Authors · 2025-11-21
>
> We sincerely appreciate the detailed feedback and constructive suggestions from the reviewers. We are pleased that they found the paper well-written and the methodology clear (3vm7, N8FK, ZEm1). Reviewers also highlighted that our work addresses an important problem with high application potential (3vm7, Bdnm) using a novel approach to the contrastive loss (3vm7, Bdnm) and thorough ablations (Bdnm, ZEm1). We have thoroughly considered all comments and concerns and have provided detailed responses below. Most notably, we have incorporated a scaling analysis on ESM2, from 8M to 650M, computed confidence intervals for all reported benchmarks, and added new benchmarks, including ProteinGym and CASP16, showcasing the strengths of our method.
>
> ---
>
> **Comment.** There is a related work which could bear including in the baseline [1]
>
> **Answer.** We propose to add the following discussion to the Related Work: “[1] incorporates structural information by using structure-extractor modules to refine the pLM’s self-attention. However, this approach is not directly comparable to ours as the Protein Structure Transformer requires explicit structural inputs, whereas SaESM operates purely on sequences.”  We cannot evaluate it on the benchmarks considered because it requires explicit structural inputs.
>
> **Comment.** The authors should explore a direct feature fusion approach. It is not clear what is left once even the dual loss is removed and is probably not the strongest form of this ablation. [2]
>
> **Answer.** Our approach assumes that pLMs only have access to sequences due to the real-world constraint that many proteins lack structures (e.g., about 50% of the human proteome is intrinsically disordered). Therefore, feature-fusion methods requiring additional modalities during inference, such as structural features, are outside our scope. While multi-modal models like ESM3 also employ feature fusion, their technique fundamentally differs from Attentional Feature Fusion.

---

### Official Review · Reviewer_3vm7 · 2025-11-04

**Soundness:** 2
**Presentation:** 3
**Contribution:** 2
**Rating:** 2
**Confidence:** 4

**Summary:**

Proteins are sequences of amino acids that fold to a 3d structure. As such, representation learning can be done on the sequence space, for example by masking and predicting parts of the sequence, or in the structure space, by designing a self-supervised task around the 3d-structure of the protein. The authors argue that protein language models trained on the sequence space lack structural knowledge and can be improved by a contrastive learning task using pretrained graph neural networks trained on the structure domain. They call this structure alignment. They use a contrastive method toe align protein language models with structure information and show that these methods perform better than sequence only methods.

**Strengths:**

- The paper is well written and the methodology is clear.
- The authors tackle an important problem of creating better protein sequence representations that account for the multimodal nature of proteins.
- The specific method of computing the loss seems novel.

**Weaknesses:**

There are several methods that combine sequence and structure including contrastive learning

e.g:
[1] CCPL: Cross Modal Contrastive Protein Learning https://arxiv.org/abs/2303.11783,

[2] S-PLM: Structure-aware Protein Language Model via Contrastive Learning between Sequence and Structure https://pubmed.ncbi.nlm.nih.gov/37609352/

[3] BioCLIP - Contrasting Sequence with Structure: Pre-training Graph Representations with PLMs https://www.biorxiv.org/content/10.1101/2023.12.01.569611v1

[4] ProTrek that performs contrastive learning across sequence structure and function https://www.biorxiv.org/content/10.1101/2024.05.30.596740v1.full.pdf)

[5] Cross-modality and self-supervised protein embedding for compound–protein affinity and contact prediction https://pmc.ncbi.nlm.nih.gov/articles/PMC9486597

The authors does not distinguish how their work differs from all this prior work and how this is significant. The evaluation also does not consider these works.

**Questions:**

What is the main difference between this work and all the prior work on structure aware protein language models. How does it compare in terms of methodology and performance?
Could you please look at the tasks covered in these papers (and many others), and make the baselines more comprehensive.

---

> ### Author Response · Authors · 2025-11-21
>
> We sincerely appreciate the detailed feedback and constructive suggestions from the reviewers. We are pleased that they found the paper well-written and the methodology clear (3vm7, N8FK, ZEm1). Reviewers also highlighted that our work addresses an important problem with high application potential (3vm7, Bdnm) using a novel approach to the contrastive loss (3vm7, Bdnm) and thorough ablations (Bdnm, ZEm1). We have thoroughly considered all comments and concerns and have provided detailed responses below. Most notably, we have incorporated a scaling analysis on ESM2, from 8M to 650M, computed confidence intervals for all reported benchmarks, and added new benchmarks, including ProteinGym and CASP16, showcasing the strengths of our method.
>
> ---
>
> **Comment.** What is the main difference between this work and all the prior work on structure aware protein language models?
>
> **Answer.** We clarify how our method differs from each category:
>  * CCPL [1] and BioCLIP [3] fundamentally differ from the proposed method as they focus on improving pGNNs, not pLMs. Notably, they maintain the pLM frozen while optimizing the pGNN, whereas our method aims to improve the pLM by injecting structural information from a frozen pre-trained pGNN. As such, evaluating them would amount to evaluating the base pLM. Furthermore, we retain the LM head and MLM task, enabling zero-shot DMS scoring.
>  * S-PLM [2] and ProTrek [4] only consider latent embedding alignment and do not incorporate a physical-level task, which our ablations show to be important (Tab. 2). Furthermore, their contrastive learning operates at the protein-level, while our latent-level task is conducted at the residue-level, enabling finer-grained structural supervision. They also do not make use of a pre-trained pGNN to extract structural features and S-PLM considers only contact maps and not structures. In comparison, our method is cheaper and produces finer-grained embeddings, which allow for a richer set of downstream tasks. Both these methods also discard the LM head, preventing comparison on ProteinGym.
>  * Cross-modality protein embedding for compound–protein affinity/contact prediction [5] focuses solely on residue token prediction and does not incorporate the latent-level and physical-level tasks that are essential components of our framework, which our ablations show to be important (Tab. 2)
>
> We propose to add the following discussion regarding [2, 4, 5] to the Related Work: “S-PLM and ProTrek primarily focus on latent embedding alignment and do not incorporate a physical-level task, which our ablation shows to be crucial (Tab. 2). Furthermore, their contrastive learning is performed at the protein-level, in contrast to our latent-level task that operates at the residue-level. Finally, [5] is limited to residue token prediction.”
>
> [1] and [3] are already addressed in the Related Work section. Unfortunately, we are unable to evaluate CCPL as its model weights are not publicly available. We indeed missed S-PLM and are currently evaluating it on the relevant benchmarks and will update you as soon as we have the results.
>
> Finally, we suspect there is a significant data leakage in ProTrek due to its use of text descriptions during training, which exposes test set properties such as GO, EC, and subcellular localization. A brief manual inspection of the test set sequences supports our suspicion that this leakage is real, which prevents a fair and direct comparison against our method.
>
> **Comment.** Could you please look at the tasks covered in these papers (and many others), and make the baselines more comprehensive.
>
> **Answer.** We've significantly extended our evaluation suite by adding 5 new downstream property prediction tasks and including confidence intervals for all of them. (1) Contact map prediction capability on CASP16, which is completely deduplicated from all model training sets. (2) Protein mutations effect prediction on ProteinGym, which offers a distinctly different evaluation approach and covers over 2 million deep mutation scanning results across 217 separate assays. (3) Enzyme catalytic efficiency, TCR-pMHC affinity, and peptide HLA-MHC affinity prediction from the xTrimoPGLM evaluation suite.

---

> > ### Author Response · Authors · 2025-12-03
> >
> > **Comparison with S-PLM**
> > We have finalized our evaluation of S-PLM on the different relevant benchmarks and updated Section 4 and Appendix D with the results. On average over xTrimoPGLM benchmarks, our method significantly outperforms S-PLM (+4.3%), with the most notable improvements for contact map prediction on CASP16, secondary structure prediction (5.4%) and stability prediction (+23%). Our conclusion on saturation of performance for the SaProt benchmark remains largely unchanged, with the exception of Human PPI prediction, where S-PLM slightly outperforms our model and the base ESM2 model, while being comparable with the other structure aligned baselines

---

### Author Response · Authors · 2025-12-03

Dear AC,

Following the ICLR decision concerning leaked identities, we decided to expand on our response to give a comprehensive overview.

Overall, reviewers found our paper well-written and the methodology clear (3vm7, N8FK, ZEm1). Reviewers also highlighted that our work addresses an important problem with high application potential (3vm7, Bdnm) using a novel approach to the contrastive loss (3vm7, Bdnm) and thorough ablations (Bdnm, ZEm1). Reviewers mentioned several concerns that we thoroughly considered and addressed as follow:
- We have incorporated a scaling analysis on ESM2, from 8M to 650M (requested by Reviewer N8FK). We also discussed in our rebuttals to Reviewer N8FK our choice not to post-train larger models (3B, 15B) due to observed performance saturation and high computational costs.
- We have added a new baseline S-PLM to all relevant downstream tasks evaluations (requested by Reviewers 3vm7 and N8FK). Reviewers also mentioned other works that we cannot compare against because either their models are not open source (e.g., CCPL, mentioned by Reviewer 3vm7) or we identified data leakage (e.g., ProTrek, see our answer to Reviewer 3vm7).
- We have added several new evaluations: (i) ProteinGym, which covers deep mutational scanning (DMS) fitness prediction. (ii) CASP16, for contact map prediction on completely deduplicated data. (iii) 5 other downstream property prediction tasks from the xTrimoPGLM or SaProt evaluation suites (including enzyme catalytic efficiency and TCR-pMHC affinity). We achieve statistically significant and substantial improvements on structure-dependent tasks, most notably a +59% increase in Precision@L on CASP16 and consistent gains on ProteinGym (+6-7% for 35M and 150M models). For the interaction-based tasks (e.g., TCR-pMHC), performance remains largely comparable to baselines, which we attribute to our method's focus on single-chain structural injection.
- For all downstream evaluations, we added 95% confidence intervals, highlighting when our improvements are statistically significant (e.g., CASP16, ProteinGym) but also identifying many tasks where all performance is saturated or overlapping. We added a discussion on potential reasons for this saturation in Appendix C.2.
- Lack of Novelty: We clarified that our contribution extends beyond standard contrastive learning by introducing a dual-task framework that captures both inter- and intra-protein structural information at the fine-grained residue level. Crucially, unlike prior structure-infused methods (e.g., S-PLM, ProTrek) that discard the language modeling head to focus solely on representation learning, our approach preserves the LM head and the MLM task. We argued that this design choice is significant as it enables zero-shot capabilities for tasks like Deep Mutational Scanning on ProteinGym for example, which are impossible for methods that function purely as residue or sequence embedding models.

The following major concerns were addressed during the discussion but led to no modifications of the paper:
- Comparison with Structure-Input and Feature Fusion Methods (e.g., SaProt): We addressed concerns regarding comparisons to SaProt and requests to explore feature fusion (e.g., Attentional Feature Fusion) by emphasizing a fundamental difference in improved applicability: these methods require explicit structural inputs (e.g., Foldseek tokens or 3D coordinates) at inference time. In contrast, our method is strictly sequence-only during inference, injecting structural knowledge solely via the contrastive loss. We argued this distinction is critical for applicability to the ~50% of the proteome that is intrinsically disordered or lacks reliable structures, and ensures our method maintains the computational efficiency of standard pLMs without the overhead of structure generation.
- Theoretical Redundancy (Anfinsen’s Principle): In response to concerns that structural information might be redundant given that sequence determines structure (Anfinsen’s principle), we argued that while the two are theoretically linked, explicit structural injection provides a complementary view that improves learning efficiency. We also argued that sequence-only models often struggle to implicitly infer complex geometric regularities required for tasks like stability prediction or contact map prediction, a claim supported by our gains on these specific downstream tasks.

We believe that we addressed the most important weaknesses the reviewers mentioned and even went beyond what they asked with additional evaluations and analysis. All modifications to the paper since the reviewing started are highlighted in blue in the attached pdf.
Finally, we believe the ICLR audience will benefit from our detailed ablations and expanded evaluations. Our inclusion of confidence intervals helps identify limitations in previous benchmarks, while our approach highlights how efficient post-training methods can deliver substantial performance improvements.

---

### Meta-Review · Area_Chair_nTMX · 2026-01-06

**Summary:**

This submission proposes a method to enhance protein language models (pLMs) with structural knowledge. The core idea involves a post-training dual-task framework: (1) a latent-level contrastive loss that aligns residue embeddings from a pLM with those from a frozen protein graph neural network (pGNN), and (2) a physical-level task predicting discrete structural tokens (e.g., from Foldseck). An auxiliary residue loss selection module is introduced to prioritize learning from reliable yet challenging residues based on a quality-filtered reference model. The method is applied to ESM2 and AMPLIFY, resulting in models named SaESM2 and SaAMPLIFY. Evaluations across multiple benchmarks (CASP16, ProteinGym, xTrimoPGLM, SaProt) show performance gains on several structure-prediction tasks (e.g., contact prediction) and in zero-shot deep mutational scanning. The authors emphasize that their approach maintains the original masked language modeling (MLM) head, enabling zero-shot capabilities without requiring structural input at inference time.

While the objective of bridging sequence and structure is valuable, the paper suffers from significant issues regarding readiness and novelty. The proposed method essentially combines existing components (CLIP-style loss, Foldseek tokens, GearNet) in a manner that is largely incremental engineering rather than a fundamental methodological innovation. The original submission appeared to be prepared in a rush, lacking critical baselines (e.g., S-PLM), essential benchmarks (e.g., ProteinGym, CASP16 were only added during rebuttal), and suffering from cluttered presentation in figures and tables. While the authors conducted extensive experiments during the rebuttal to fill these gaps, the initial lack of rigor and the mixed results on downstream functional tasks (where gains are often marginal or statistically insignificant compared to unaligned baselines) suggest the work is not yet polished enough for ICLR.  The presentation remains cluttered, and the visual quality of the charts reflects the hasty preparation. The paper requires a thorough revision, integration of the new results into a coherent narrative, and better polishing before it meets the bar for a top-tier venue.

**Reviewer Concerns:**

The reviewers identified several fundamental weaknesses that persist despite the rebuttal efforts:

1.  **Limited Novelty:** Reviewers (3vm7, N8FK, ZEm1) questioned the methodological contribution, noting that contrastive learning between sequence and structure is a standard technique. The distinction of operating at the "residue-level" is viewed as an incremental variation rather than a significant breakthrough.
2.  **Incompleteness of Original Submission:** Multiple reviewers (3vm7, N8FK) flagged missing baselines (e.g., S-PLM, BioCLIP) and insufficient evaluations. The authors had to add substantial amounts of data (ProteinGym, CASP16, Scaling analysis) during the rebuttal, confirming the paper was submitted prematurely.
3.  **Marginal Improvements:** Reviewers noted that outside of direct structural tasks (like contact prediction), the improvements on functional property prediction are often marginal or exhibit overlapping confidence intervals with baselines.
4.  **Presentation Quality:** Reviewer N8FK highlighted that the paper was hard to read, with confusing terminology (e.g., the non-standard use of "inter-protein") and messy visualizations, indicating a lack of polish.

**Reviewer Scores:**

Initial scores reflected significant reservations: **3vm7: 2 (Reject), N8FK: 2 (Reject), ZEm1: 6 (Marginally Accept), Bdnm: 6 (Marginally Accept)**. The authors' **rebuttal was diligent**, adding evaluations (ProteinGym, CASP16), confidence intervals, scaling analysis, and a new baseline (S-PLM). This addressed some technical concerns, leading **N8FK to indicate an intent to raise their score** and **ZEm1 to explicitly increase their rating**. However, the fundamental concerns regarding the **incremental nature of the contribution and the paper's foundational rigor** were not fully dispelled. Reviewer 3vm7's score remained unchanged, indicating persistent doubts about the work's significance relative to the existing literature.

---

### Decision · Program_Chairs · 2026-01-26

Reject